# Implantation initiation of self-assembled embryo-like structures generated using three types of mouse blastocyst-derived stem cells

Shaopeng Zhang [1], Tianzhi Chen[1], Naixin Chen[1], Dengfeng Gao[1], Bingbo Shi [1], Shuangbo Kong[2], Rachel Claire West[3], Ye Yuan[3], Minglei Zhi [1], Qingqing Wei[1], Jinzhu Xiang[1], Haiyuan Mu[1], Liang Yue[1], Xiaohua Lei[4], Xuepeng Wang[4], Liang Zhong[5], Hui Liang[1], Suying Cao[6], Juan Carlos Izpisua Belmonte[7], Haibin Wang[2] & Jianyong Han [1,8]

Spatially ordered embryo-like structures self-assembled from blastocyst-derived stem cells can be generated to mimic embryogenesis in vitro. However, the assembly system and developmental potential of such structures needs to be further studied. Here, we devise a nonadherent-suspension-shaking system to generate self-assembled embryo-like structures (ETX-embryoids) using mouse embryonic, trophoblast and extra-embryonic endoderm stem cells. When cultured together, the three cell types aggregate and sort into lineage-specific compartments. Signaling among these compartments results in molecular and morphogenic events that closely mimic those observed in wild-type embryos. These ETX-embryoids exhibit lumenogenesis, asymmetric patterns of gene expression for markers of mesoderm and primordial germ cell precursors, and formation of anterior visceral endoderm-like tissues. After transplantation into the pseudopregnant mouse uterus, ETX-embryoids efficiently initiate implantation and trigger the formation of decidual tissues. The ability of the three cell types to self-assemble into an embryo-like structure in vitro provides a powerful model system for studying embryogenesis.

---

[1] State Key Laboratory of Agrobiotechnology, College of Biological Sciences, China Agricultural University, Beijing 10094, China. [2] Fujian Provincial Key Laboratory of Reproductive Health Research, Medical College of Xiamen University, Xiamen, Fujian 361102, China. [3] Colorado Center for Reproductive Medicine, Lone Tree, CO 80124, USA. [4] State Key Laboratory of Stem Cell and Reproductive Biology, Institute of Zoology, Chinese Academy of Sciences, Beijing 100101, China. [5] CAS Key Laboratory of Pathogenic Microbiology and Immunology, Institute of Microbiology, Chinese Academy of Sciences, Beijing 100101, China. [6] Animal Science and Technology College, Beijing University of Agriculture, Beijing 102206, China. [7] Salk Institute for Biological Studies, La Jolla, CA 92037, USA. [8] Advanced Innovation Center for Food Nutrition and Human Health, China Agricultural University, Beijing 100083, China. These authors contributed equally: Shaopeng Zhang, Tianzhi Chen. Correspondence and requests for materials should be addressed to J.H. (email: hanjy@cau.edu.cn)

The mammalian zygote undergoes a series of changes, including zygotic genome activation and lineage specification, that are each critical for generating a blastocyst. The blastocyst is comprised of an inner cell mass (ICM) within the trophectoderm (TE), with the ICM including the epiblast (EPI), and primitive endoderm (PE)[1,2]. During implantation, the blastocyst undergoes a morphogenetic transformation in which the original vesicular structure is reorganized into an elongated structure at E6.5. This elongated structure is made up of: (1) the ectoplacental cone, (2) the EPI, (3) the extra-embryonic ectoderm (ExE), (4) a layer of visceral endoderm (VE) that envelopes both the EPI and ExE, and (5) the parietal yolk sac, Reichert's membrane, and trophoblast giant cell (TGC) layer, which together surround the entire conceptus[3-6]. During gastrulation (i.e., the formation of a gastrula from a blastula), communication between these embryonic tissues causes the EPI cells to polarize, adopt a rosette-like configuration, and undergo lumenogenesis. This is followed by development of the trophectoderm into the ExE, which forms a second cavity[7,8]. Both the embryonic and extra-embryonic cavities unite to form a single pro-amniotic cavity, and the embryo breaks symmetry to initiate the specification of mesoderm and primordial germ cells[9]. The VE is a particularly important source of signals for embryonic patterning[5]. Precursor cells of the anterior VE (AVE) arise at the distal tip of the embryo (termed the distal VE, DVE) and then migrate to the anterior side of the embryo. The AVE is crucial for anterior-posterior patterning, as it is a source of antagonists for posteriorizing signals, such as Nodal and Wnt[10-12]. By the end of gastrulation, the three primary germ layers have been formed, including the ectoderm, mesoderm and definitive endoderm, from which all fetal tissues will develop.

Stem cells have been derived from the three cell lineages of the mouse blastocysts, namely, embryonic stem cells (ESCs) from the EPI[13], extra-embryonic endoderm stem cells (XENCs) from the PE[14], and trophoblast stem cells (TSCs) from the TE[15]. Each of these stem cell types can be maintained indefinitely in culture. ESCs can differentiate into cells from all three germ layers[13,16], and can be induced to form embryoid bodies (EBs) or micropatterned colonies. These are valuable tools for studying embryonic development, but EBs do not fully recapitulate the spatial-temporal events of embryogenesis, nor do they acquire the cellular architecture of a post-implantation embryo[17-20]. Recently, ESCs and TSCs were combined in a three-dimensional (3D)-scaffold to produce ETS-embryoids that undergo embryogenic process similar to normal embryogenesis[9]. However, these embryo-like structures lack PE-derived cells, which may play critical roles during later stages of embryogenesis[5,21]. Here, we mimic embryogenesis in vitro by culturing together the three types of blastocyst-derived stem cells (ESCs, TSCs, and XENCs; we refer to this combination as ETX) using a nonadherent-suspension-shaking system. We hypothesize that if these cell types were cultured together under suitable conditions, they would engage in both homo- and heterotypic interactions necessary for embryo formation. Indeed, interactions between these stem cells in this suspension system recapitulate many of the molecular and morphogenic events of early mouse embryogenesis, resulting in the generation of what we call ETX-embryoids.

## Results

**Forming self-assembled structures under nonadherent-suspension-shaking culture system.** Individual cells in tissues and organs are able to recognize, adhere to, and communicate with each other through binding between cell surface molecules. The three types of blastocyst-derived stem cells (ESCs, TSCs, and XENCs) are no exception, as they each express lineage-specific cell surface proteins[22,23]. We hypothesized that if the three blastocyst-specific stem cell types were cultured together under suitable conditions they would be able to aggregate, sort into specific cellular compartments, and then recapitulate the signaling and morphogenic events necessary for mouse embryogenesis. To establish these conditions, we used ESC and TSC reporter lines (DsRed-ESCs and EGFP-TSCs) (Supplementary Fig. 1a, b) to trace their spatial location in aggregated structures, and compared different methods, including hanging drop culture[24], microporous culture[25,26], microinjection method, and nonadherent-suspension-shaking culture (Supplementary Fig. 1c–f). The last method was proved to be the most suitable for promoting cell–cell contacts, cell–cell self-recognition, and cellular aggregation into ordered structures. Thus, the nonadherent-suspension-shaking culture method was used for further studies.

To investigate how different types of stem cells assemble together, we designed an experimental strategy (Fig. 1a). We first examined combinations of two cell types. When ESCs and TSCs were cultured together, they formed a cylindrical structure similar to the recently reported ETS-embryoid[9] (Supplementary Fig. 1g, h). The TSC/XENC combination did not form a specific structure, whereas ESCs and XENCs self-assembled into a structure, in which a XENC-derived cell layer surrounded an ESC sphere (EXE-embryoid) (Fig. 1b and Supplementary Fig. 1i). A similar to result of EPI-explants in vitro cultured from the EPI-part wrapped with the VE of E5 embryos[27,28] (Fig. 1c).

When the three stem cell types were combined, they generated embryonic-like structures (ETX-embryoids) that had spatial relationships between the cell types that resembled post-implantation embryos. In these ETX-embryoids, ESC-derived cells were on one side, TSC-derived cells were on the other side, and XENC-derived cells enveloped the ESC- and TSC-derived structures (Fig. 1d, e and Supplementary Fig. 1j–m). Localization of the ESC-, TSC- and XENC-derived cells was further confirmed via immunostaining, using antibodies specific for the lineage markers Gata4, Oct4, Cdx2, and Eomes (Fig. 1b–e and Supplementary Fig. 1h, i, k–m). Eomes is expressed in the ExE and embryonic VE, and is robustly induced at the onset of gastrulation[29]. Labeling wild-type embryos for Eomes showed the ExE developed from the TE, and the VE developed from the PE (Fig. 1f, g). Lineage marker gene expression levels for single cells isolated from ESC-, TSC-, and XENC-specific compartments confirmed their spatial localization in ETX-embryoids at 84 h (Fig. 1h). The volume of ESC and TSC compartments in 72 h ETX-embryoids was comparable to that seen for E5.75 embryos (Fig. 1i).

We found that the initial number of cells, the cell ratios, and the medium composition affected the efficiency of formation for these reconstructed embryos (Supplementary Fig. 2a, e–h). When we visually assessed ETS-embryoids, we identified both regular and irregular structures (Supplementary Fig. 2b, c). We observed higher efficiency (35%) and better morphology of ETS-embryoids using a mixture of $1 \times 10^5$ ESCs and $1 \times 10^5$ TSCs in a 3.5-cm dish when shaken at 60 revolutions per minute (Supplementary Fig. 2a, d). A previous study reported that ETS-embryoids must be cultured in a medium that includes FGF4 for TSC self-renewal and proliferation[9,15]. Here, we found that the addition of small molecules, such as LIF, 2i (CHIR99021 and PD0325901), and FGF4, which inhibit ESC and TSC differentiation respectively, affected the formation of the reconstructed embryos (Supplementary Fig. 2e–h). Therefore, we did not include these factors in our culture system for subsequent self-assembly experiments.

We next examined the reproducibility by which these EXE-, ETS-, and ETX-embryoids could be reconstructed. The EXE-embryoids were most efficiently assembled, with > 90% exhibiting

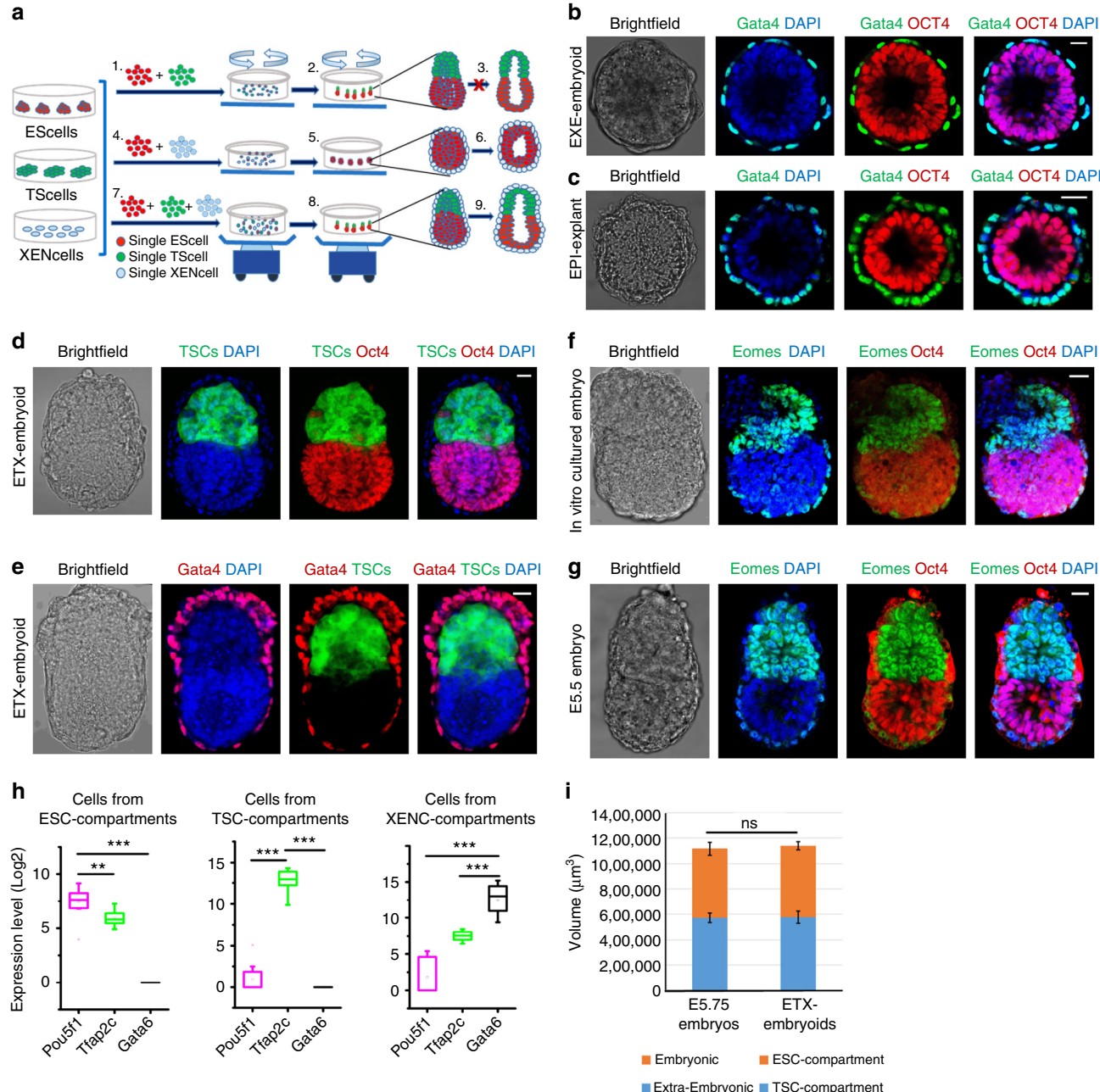

**Fig. 1** Self-assembly of different combinations of distinct stem cells into reconstructed embryoids. **a** Schematic of the protocol to generate self-assembled embryoids. Single-cell populations of ESCs and TSCs were mixed and transferred into a dish containing reconstructed embryo medium on an orbital and horizontal shaker (1), and embryo-like structures (ETS-embryoids) emerged within 24 h (2) but did not undergo lumenogenesis in further culture (3). Similarly, a combination of ESCs and XENCs produced EXE-embryoids within 24 h (4, 5), and these embryoids formed cavities (6). ETX-embryoids were generated within 36 h by combination of the three types of stem cells (7, 8), and these embryos also formed cavities (9). **b**, **c** Immunostaining of EXE-embryoids and EPI-explants after culture of 60 and 24 h: DAPI, blue; Gata4, green; and Oct4, red. $n = 45$ EXE-embryoids; $n = 45$ EPI-explants. **d**, **e** Immunostaining of ETX-embryoids assembled after culture of 72 h from wild-type XENCs and ESCs, EGFP-TSCs: DAPI, blue; TSC-derived tissues, green; Oct4 and Gata4, red. $n = 30$ ETX-embryoids. **f**, **g** Immunostaining of embryos cultured in vitro for 60 h from the E4.5 blastocyst and E5.5 embryos: Oct4, red; Eomes, green; and DAPI, blue. $n = 20$ embryos. **h** Box-plots of expression levels of *Pou5f1*, *Tfap2c* and *Gata6* in cells from ESC-, TSC-, and XENC-compartments, respectively, of the ETX-embryoids. Each cell type is shown in a separate color. The boxed region represents the middle 50% of expression values, the bar in the boxed region indicates the median values, and the whiskers indicate the maximum and minimum values. Cells with outlying expression values are depicted as dots. A background of Ct = 24 was used to obtain expression levels. Two-tailed Student's *t*-test, $n = 11, 11$, and 11 single cells from ESC-, TSC-, and XENC-compartments, respectively; **$P < 0.01$, ***$P < 0.001$. **i** Mean tissue volumes of embryonic and extra-embryonic structures are similar between ETX-embryoids after 84 h and wild-type E5.75 embryos. Two-tailed Student's *t*-test, $n = 34$ (ETX-embryoids) and 10 (E5.75 embryos); ns, not significant, source data are provided as the Source Data file. Scale bar, 20 μm (**b–g**). Columns are means ± s.e.m. Experiments were repeated at least three times (**b–g**).

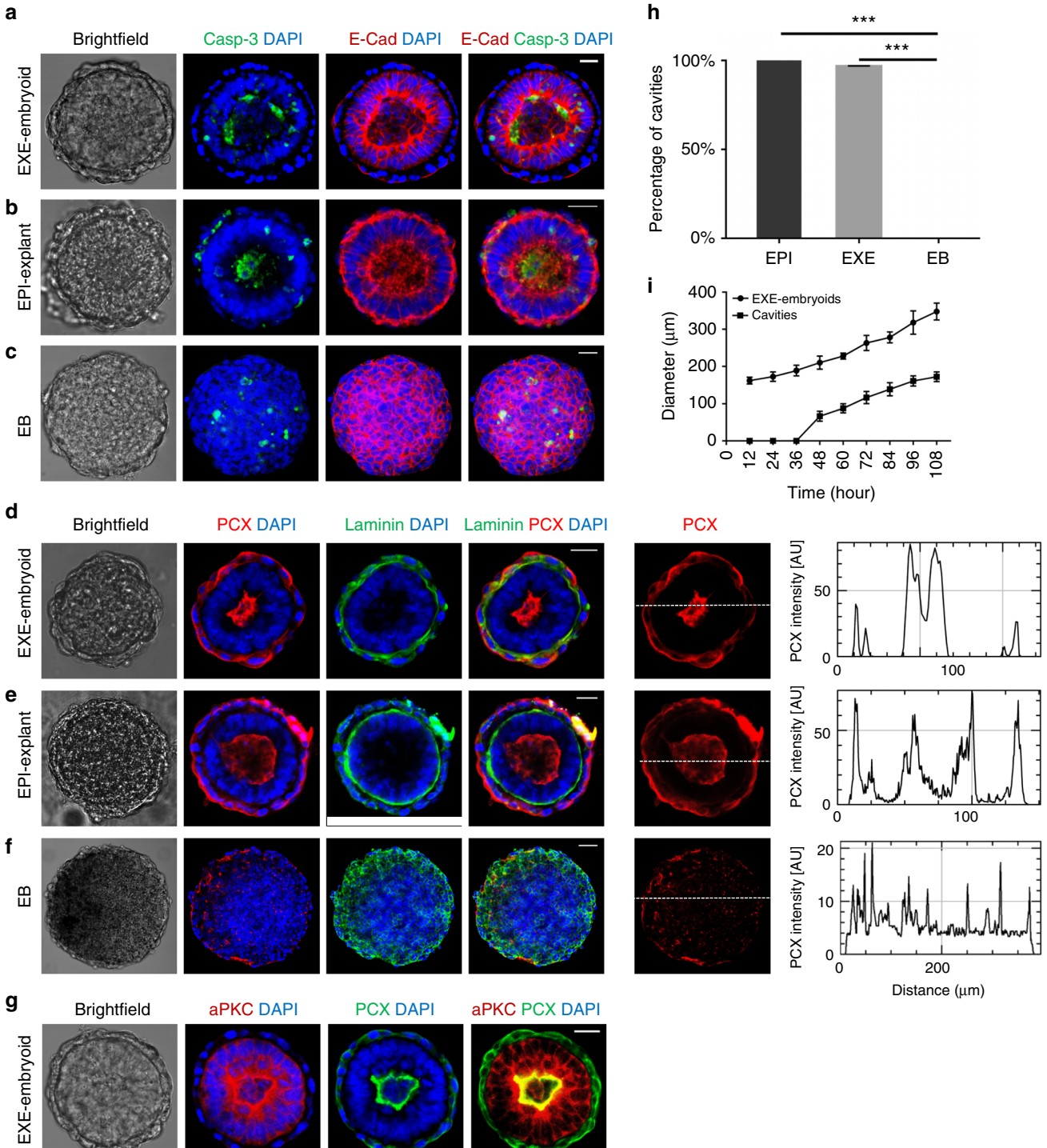

**Fig. 2** Lumenogenesis in EXE-embryoids. **a** EXE-embryoids stained to reveal cleaved caspase-3, green; E-Cadherin, red; and DAPI, blue. $n = 20$ EXE-embryoids. **b** EPI-explants stained to reveal cleaved caspase-3, green; E-Cadherin, red; and DAPI, blue. $n = 10$ EPI-explants. **c** EBs stained to reveal cleaved caspase-3, green; E-Cadherin, red; and DAPI, blue. $n = 30$ EBs. **d** EXE-embryoid shows the cavity and Laminin niche via immunostaining: PCX, red; Laminin, green; and DAPI, blue. $n = 30$ EXE-embryoids. The right-most panel shows the intensity scan of PCX along a white line drawn at the middle z-plane of the EXE-embryoid; the presence of the cavity is indicated by two strong peaks in the intensity profile. **e** EPI-explants show the cavity and laminin niche via immunostaining: PCX, red; Laminin, green; and DAPI, blue. $n = 30$ EPI-explants. The right-most panel shows the intensity scan of PCX along a white line drawn at the middle z-plane of the EPI-explants, indicating cavity formation. **f** EBs cannot form a cavity or Laminin niche: PCX, red; Laminin, green; and DAPI, blue. $n = 30$ EBs. The right-most panel shows the intensity scan of PCX along a white line drawn at the middle z-plane of the EBs, indicating no cavity formation. **g** Localization of PCX and aPKC in EXE-embryoids show cavity, aPKC marked apical side of cells. PCX, green; aPKC, red; and DAPI, blue. $n = 20$ EXE-embryoids. **h** Percentage of cavity formation in EXE-embryoids, EPI-explants, and EBs. Two-tailed Student's $t$-test; means ± s.e.m. Three experiments. ***$P < 0.001$. **i** The mean external diameters of the EXE-embryoids and the mean internal diameters of the cavities in the EXE-embryoids enlarged along with time of in vitro culture. $n = 20$ EXE-embryoids at each time points. Means ± s.d. Scale bar, 20 μm (**a–e**, **g**), 50 μm (**f**). Experiments were repeated at least three times with similar results (**a–g**). Source data of **h** and **i** are provided as the Source Data file

regular structures (i.e., the ESC compartment surrounded by XENC-derived cells) (Fig. 1b and Supplementary Fig. 1i). For ETX-embryoids, an initial ratio of $1 \times 10^5$ ESCs, $1 \times 10^5$ TSCs, and $2 \times 10^5$ XENCs in a 3.5-cm dish generated the highest percentage (~23%) of regular ETX-embryoids (Supplementary Fig. 3a–e).

Altogether, these results demonstrate that in vitro self-recognition and self-assembly of the three types of blastocyst-derived stem cells can lead to the formation of embryo-like structures in which the spatial distributions of the stem cell-derived tissues resemble the distribution of the ExE, EPI and VE in E5.5 embryos.

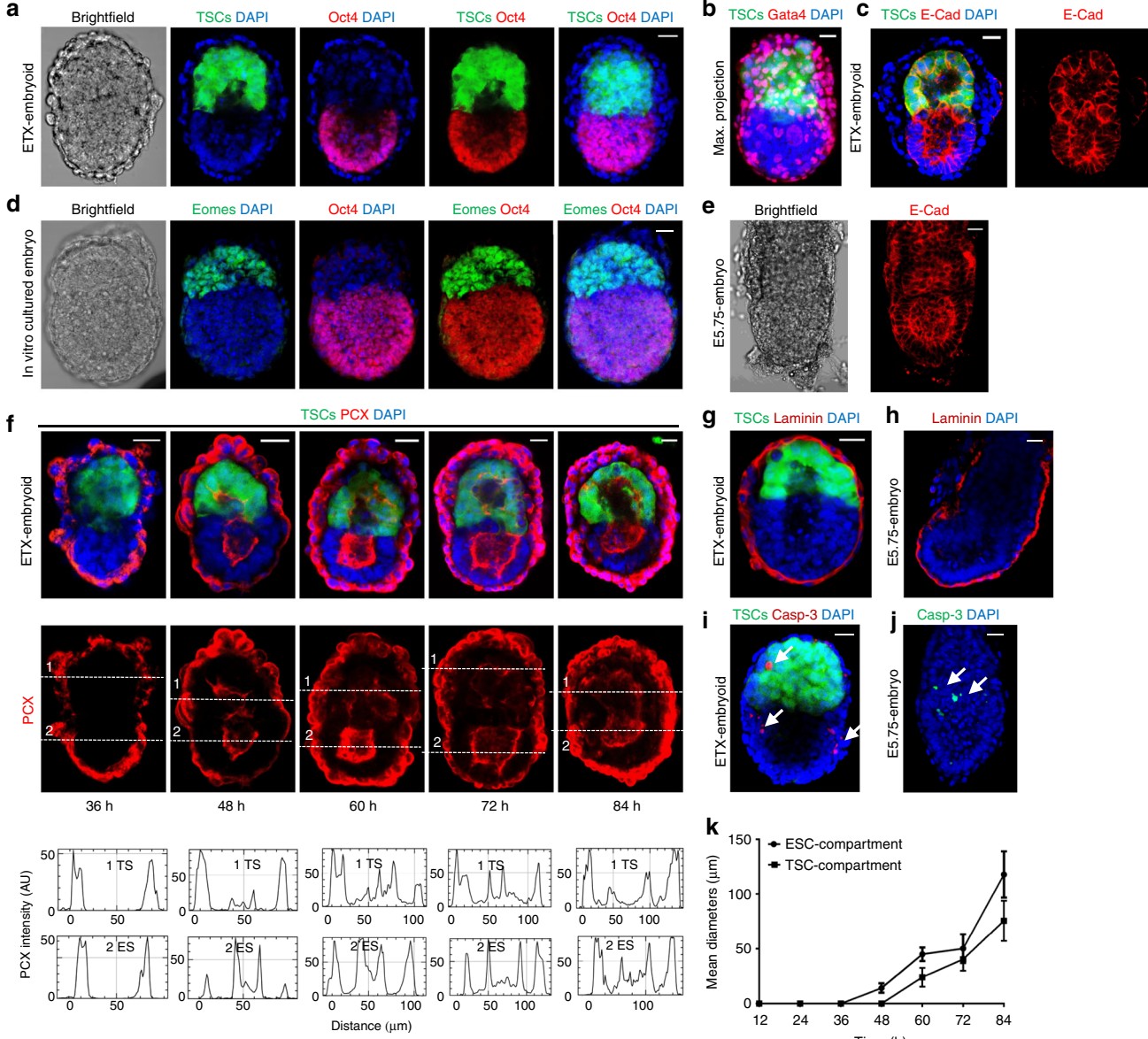

**Fig. 3** Pro-amniotic cavity formation in ETX-embryoids. **a** Localization of ESC-, TSC-, and XENC-derived tissues in ETX-embryoids at 84 h. The green cells are from EGFP-TSCs, Oct4 (red) indicates ESC-derived tissues, and DAPI only (blue) cells in the merged image (the right-most panel) show XENC-derived tissues. n = 40 ETX-embryoids. **b** Representative 3D max. projection image of ETX-embryoids reveals XENC-derived tissues via immunostaining of Gata4 (red). n = 10 ETX-embryoids. **c** Immunostaining of E-cadherin shows the cavity in ETX-embryoids. E-cadherin (red); TSCs (green) and DAPI (blue). n = 20 ETX-embryoids. **d** Immunostaining shows the embryonic and extra-embryonic parts of wild-type embryos cultured for 60 h in vitro. Oct4 (red); Eomes (green); and DAPI (blue). n = 11 embryos. **e** Immunostaining of E-cadherin (red) in wild-type E5.75 embryos. n = 10 embryos. **f** Representative ETX-embryoids after in vitro culture for 36, 48, 60, 72, and 84 h during the progression of cavitation. No cavities in the ESC compartment and TS compartment at 36 h, PCX accumulation along the apical side of cells in the ESC compartment by 48 and 60 h and then in TSC compartment by 72 h. PCX lined a single united large common cavity in ETX-embryoids by 84 h. The PCX fluorescence intensity was indicated in the lower panel. Staining indicates PCX, red; DAPI, blue. **g, h** Immunostaining shows Laminin niches (red) in ETX-embryoids (**g**). These niches are similar to those in E6.5 wild-type embryos (**h**). n = 10 ETX-embryoids and n = 10 E5.75 wild-type embryos. **i, j** ETX-embryoid and E5.75 embryo stained for cleaved caspase-3 (red, arrows) and the nuclei are stained with DAPI (blue). n = 18 ETX-embryoids and 8 wild-type embryos. **k** The mean diameters of the cavities in ESC compartments and TSC compartments of the ETX-embryoids enlarged along with time of in vitro culture. n = 7 ETX-embryoids at least at each time points. Means ± s.d. Source data are provided as the Source Data file. Scale bar, 20 μm. ETX-embryoid experiments were repeated at least three times (**a**, **b**, **c**, **f**, **g**, **i**), and wild-type embryo experiments were repeated at least two times (**d**, **e**, **h**, **j**)

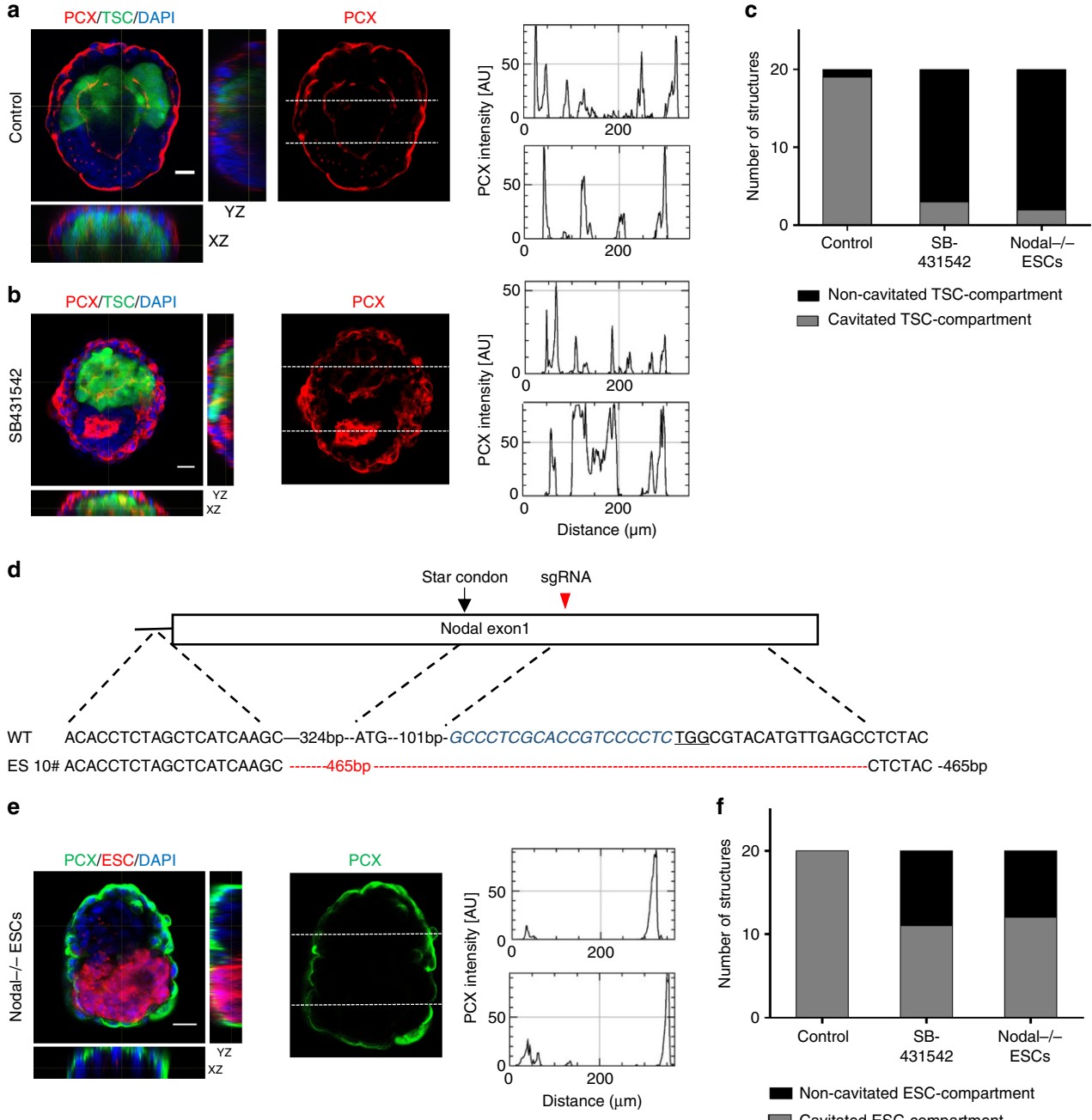

**Fig. 4** Inhibition of Nodal signaling affects cavitation of the TSC compartments in ETX-embryoids. **a, b** ETX-embryoids culutred in control condition and in 10 μM SB431542 for 84 h, respectively. PCX staining (red) highlight cavity formation in both ESC- and TSC compartments (left panel), and fluorescence traces is of PCX intensity along region indicated by the dotted lines in the ESC- and TSC compartments (right panel). Orthogonal *xz* and *yz* views are shown. *n* = 20 ETX-embryoids per group. **c** Quantification show the number of ETX-embryoids with cavitated TSC compartments after 72 h in culture in control, SB431542 and *Nodal*[−/−] ESC conditions. *n* = 20 ETX-embryoids per group. **d** Schematic diagram of gene editing at the *Nodal* loci. Wild-type sequence is shown at the top of the targeting sequence. sgRNA sequences are shown in blue. Protospacer adjacent moti (PAM) sequences are underlined. Deletions are highlighted in red. WT: wild-type; deletion "−". **e** ETX-embryoids assembled from *Nodal*[−/−] ESCs for 84 h. PCX staining (green) indicates no cavity formation in both ESC- and TSC compartments (left panel), and fluorescence trace is of PCX intensity along region indicated by the dotted lines in the ESC- and TSC compartments (right panel). **f** Quantification of ETX-embryoids with cavitated ESC compartments after 72 h in culture in control, SB431542 and *Nodal*[−/−] ESC conditions. *n* = 20 ETX-embryoids per group. Scale bar, 20 μm (**a**, **b**, **e**). Experiments were repeated at least two times with similar results. Source data of **c** and **f** are provided as the Source Data file

**Lumenogenesis in EXE-embryoids.** Next, we determined whether the reconstructed embryoids develop in a progressive sequence of spatial and temporal morphogenetic steps consistent with normal embryogenesis and examined the occurrence of events such as polarization, lumenogenesis, asymmetrical gene expression, mesoderm and PGC-like cell specification, and DVE/AVE-like tissue formation[4,5].

We first examined the development of EXE-embryoids, which initially formed after 12 h of shaking culture (Supplementary Fig. 4a). Labeling EXE-embryoids for Podocalyxin (PCX), E-

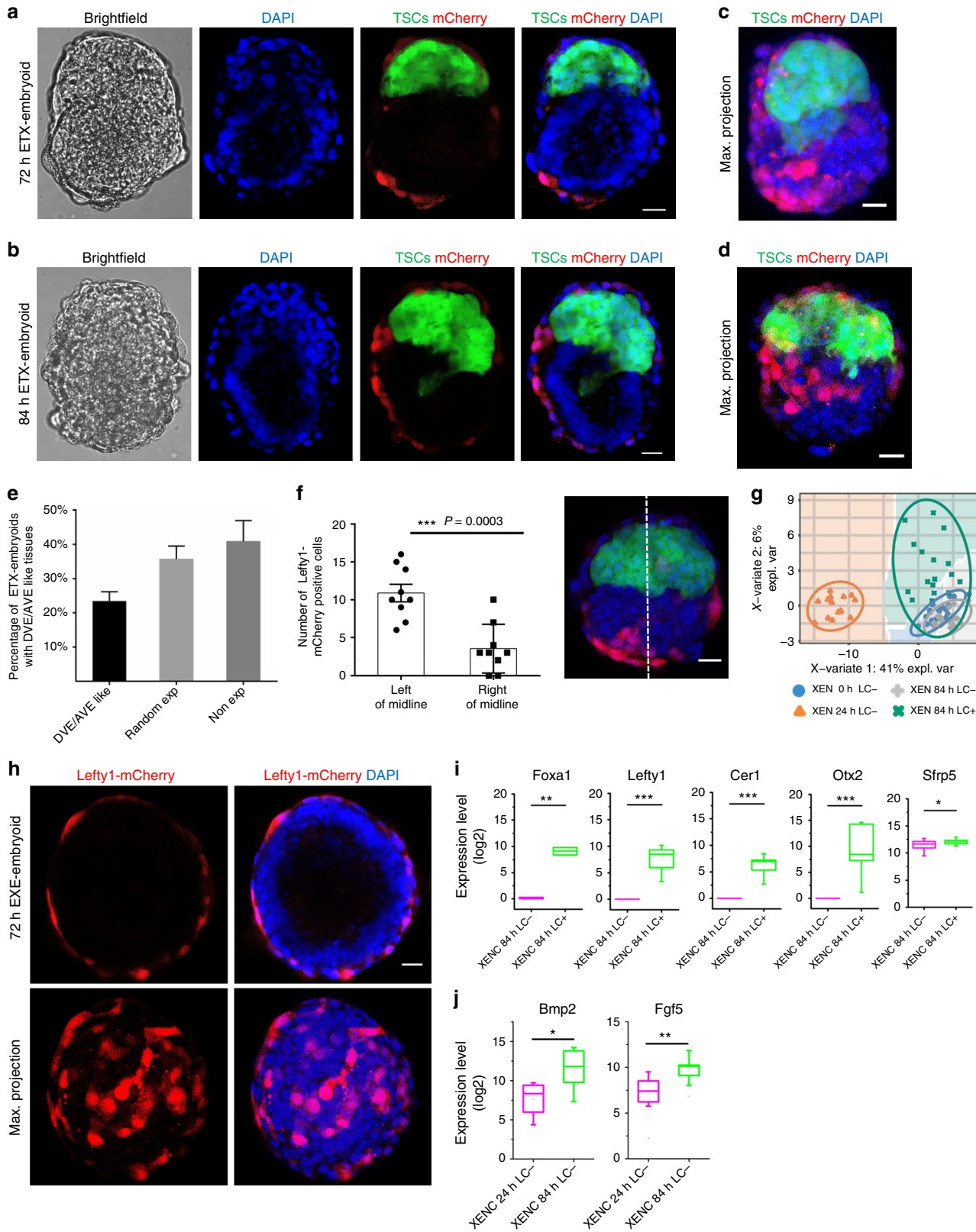

Cadherin and aPKC, which accumulate on the apical sides of EPI cells during natural pro-amniotic cavity formation[8,9], we detected cavity formation at 48 h (Fig. 2a, d, g, h, i). This lumen or cavity enlarged with prolonged culture time (Fig. 2i and Supplementary

Fig. 4b, c). Localization of the cell adhesion protein E-Cadherin revealed that the nuclei in the ESC compartment were aligned with the cavity. These morphogenetic steps are similar to those seen during the development of EPI-explants[27,28] (Fig. 2b, e).

**Fig. 5** Asymmetric Lefty1 expression in ETX-embryoids. **a**, **b** Representative ETX-embryoid assembled with *Lefty1*-mCherry XENCs, in which DVE-like tissues (red) formed at 72 and 84 h. $n = 10$ ETX-embryoids per group, three experiments. **c**, **d** Representative 3D max. projection view of ETX-embryoid with asymmetric *Lefty1*-mCherry expression (DVE-like tissues and AVE-like tissues). **e** Frequency of ETX-embryoids with DVE/AVE-like tissues. $n = 50$ ETX-embryoids per group, three experiments. Columns are means ± s.e.m. **f** Numbers of *Lefty1*-mCherry positive XEN-derived cells distributed on the left and right of midline. White line indicates the midline of the ETX-embryoids; $n = 9$ ETX-embryoids, two experiments. Two-tailed Student's *t*-test, ***$P < 0.001$. Columns are means ± s.e.m. **g** Colors and shapes are assigned to the different XEN cell groups isolated at 0, 24, and 84 h from the ETX-embryoids. The *x*-axis and *y*-axis standed for the explained variance in a PLS-DA plot. The ellipse indicates the group membership for each sample and the confidence level of ellipse is 0.95. The colored background indicates areas of each cell group. $n = 22$ (0 h), 16 (24 h), 17 (84 h, LC-) and 19 (84 h, LC +) single cells. **h** Representative EXE-embryoids, built from *Lefty1*-mCherry XENCs and cultured for 72 h, show the distribution of *Lefty1*-mCherry positive cells in a random manner. $n = 10$ EXE-embryoids per group, three experiments. **i** Box-plots of DVE/AVE markers expression levels. Single-cell qPCR assay revealed that DVE/AVE marker genes were upregulated in the mCherry positive cells compared with mCherry negative cells. **j** Box-plots of posterior markers of VE expression levels. Single-cell qPCR assay revealed that posterior marker genes were upregulated in the mCherry negative cells at 24 h compared with mCherry negative cells at 84 h. Box-plots are generated as described in Fig. 1h. A background of Ct = 26 was used to obtain expression levels. $n = 16$ single cells (24 h, LC-) and 19 (84 h, LC-), three experiments. Two-tailed Student's *t*-test, *$P < 0.05$, **$P < 0.01$, ***$P < 0.001$. Scale bar, 20 μm (**a**–**g**). LC, *Lefty1*-mCherry. Source data of **e**, **f** are provided as the Source Data file

However, we did not observe similar experimental phenomena in EBs (Fig. 2c, f). Using the previously determined optimal number of ESCs and XENCs (ESCs, $1 \times 10^5$ and XENCs, $1 \times 10^5$) and ESCs alone ($2 \times 10^5$) as controls, we found a higher number of EXE-embryoids formed compared with EBs, resulting in EBs that were larger in volume than EXE-embryoids. The average volume of EXE-embryoids at 60 h was comparable to the volume of EPI-explants cultured in vitro for 24 h (Supplementary Fig. 4d–h).

For wild-type embryos at the implantation stage of development, Laminins secreted primarily by the VE create a basal membrane niche, which helps establish EPI tissue polarity[8,30]. Immunostaining for Laminins revealed that EXE-embryoids and EPI-explants, but not EBs, were enveloped by a basal membrane (Fig. 2d–f), suggesting that Laminins play a role in the polarity and cavitation of EXE-embryoids. *Lamc1* encodes for the Laminin γ1 subunit and is necessary for basement membrane formation[31,32]. To test the role of Laminins in the formation of EXE-embryoids, we derived *Lamc1* knockout (*Lamc1*[-/-]) ESCs and XENCs to generate EXE-embryoids (Supplementary Fig. 5a). Loss of lamc1 from ESCs or XENCs did not affect the formation and cavitation of EXE-embryoids. However, EXE-embryoids were not formed when *Lamc1*[-/-] ESCs and *Lamc1*[-/-] XENCs were used (Supplementary Fig. 5b–f). This suggests that the basal membrane created by Laminins plays an important role in the development of reconstructed embryos. This is consistent with the role of Laminins in providing a signal for EPI reorganization[8,31].

We also examined cleaved caspase-3 expression in EXE-embryoids assembled from wild-type XENCs and *p53*[-/-] ESCs (Supplementary Fig. 5g, h), as well as apoptosis events in EXE-embryoids, EPI-explants and EBs (Fig. 2a–c). We found that apoptosis was not required for lumen formation, as previously reported for developing embryos[8]. Taken together, these results indicate that ESCs and XENCs can self-assemble in vitro into structures that mimic embryogenesis, including the formation of a basal membrane, the induction of cellular polarity and lumenogenesis.

**Pro-amniotic cavity formation during ETX-embryogenesis**. We generated ETX-embryoids using EGFP-TSCs and selected regular ETX-embryoids for these experiments (Fig. 3a, b and Supplementary Fig. 3d, e). Immunostaining for Oct4 indicated that ETX-embryoids at 84 h were similar to E4.5 embryos that had been cultured in vitro for 60 h. For both types of embryos, their centers lacked DAPI fluorescence, suggesting cavity formation (Fig. 3a, d and Supplementary Fig. 6a). To obtain insight into the polarization and cavitation of ETX-embryoids, we examined the distribution of E-Cadherin in ETX-embryoids at different developmental time points, and compared this to E5.75 embryos

(Fig. 3c, e and Supplementary Fig. 6b). To further confirm cavity formation, we labeled embryos for PCX, which accumulated along the apical side of cells in the ESC compartment by 60 h, and in the TSC compartment by 72 h. By 84 h, PCX lined a central common cavity. These results suggested that a small cavity could be first formed in ESC compartments and then in TSC compartments, and later, these cavities united to form a single large common cavity in ETX-embryoids (Fig. 3f). This was also supported by analyzing PCX distribution in the ESC and TSC compartments of ETX-embryoids as the cavities progressively enlarged (Fig. 3k and Supplementary Fig. 6c). Lumenogenesis in ETX-embryoids is similar to pro-amniotic cavity formation that occurs during natural embryogenesis[5,8].

Similar immunohistochemical analyses revealed that ETS-embryoids failed to acquire morphological features under the same condition (Supplementary Fig. 7a, b). This suggests that interactions between the ESC and TSC compartments are not sufficient to initiate cavitation in ETS-embryoids. This failure of cavity formation may be due to lack of basal membrane niche created by Laminins[8,9]. Thus, we further examined the distribution of Laminins in ETS-embryoids, and found that secreted Laminins were not sufficient to assemble a basal membrane that enveloped the reconstructed embryos. In 84 h ETX-embryoids, the distribution of Laminins were similar to those seen in wild-type embryos at E5.75 (Fig. 3g, h and Supplementary Fig. 7c). Further, caspase-3 localization indicated that apoptotic cells were distributed throughout the lineages and not exclusively at the center of the ESC- or TSC-derived tissues (Fig. 3i, j), suggesting that apoptosis is not essential for cavity formation[8].

**ETX-embryogenesis requires Nodal signaling**. In wild-type embryos, Nodal signaling is required throughout early post-implantation stages[33–35], and inhibition of the Nodal pathway in ETS-embryoids affects the development and cavitation of the TSC compartment[9]. To examine the role of Nodal signaling during ETX-embryogenesis, we generated ETX-embryoids in the presence of the TGF-beta receptor inhibitor SB431542, which was added when the cells were first mixed under culture conditions. Inhibition of *Nodal* signaling affected the cavity formation of the TSC compartment in ETX-embryoids (Fig. 4a–c). To further confirm the role of Nodal signaling, we designed sgRNAs to knockout Nodal in ESCs (*Nodal*[-/-] ESCs) (Fig. 4d). These cells were then used to generate ETX-embryoids. In 90% of ETX-embryoids assembled using *Nodal*[-/-] ESCs, the TSC compartment failed to cavitate, consistent with the SB431542 results. In these experiments, small effects on the ESC compartment (e.g., smaller cavities) were sometimes observed (Fig. 4c, e, f). Altogether, these experiments suggest that Nodal signaling from the ESC

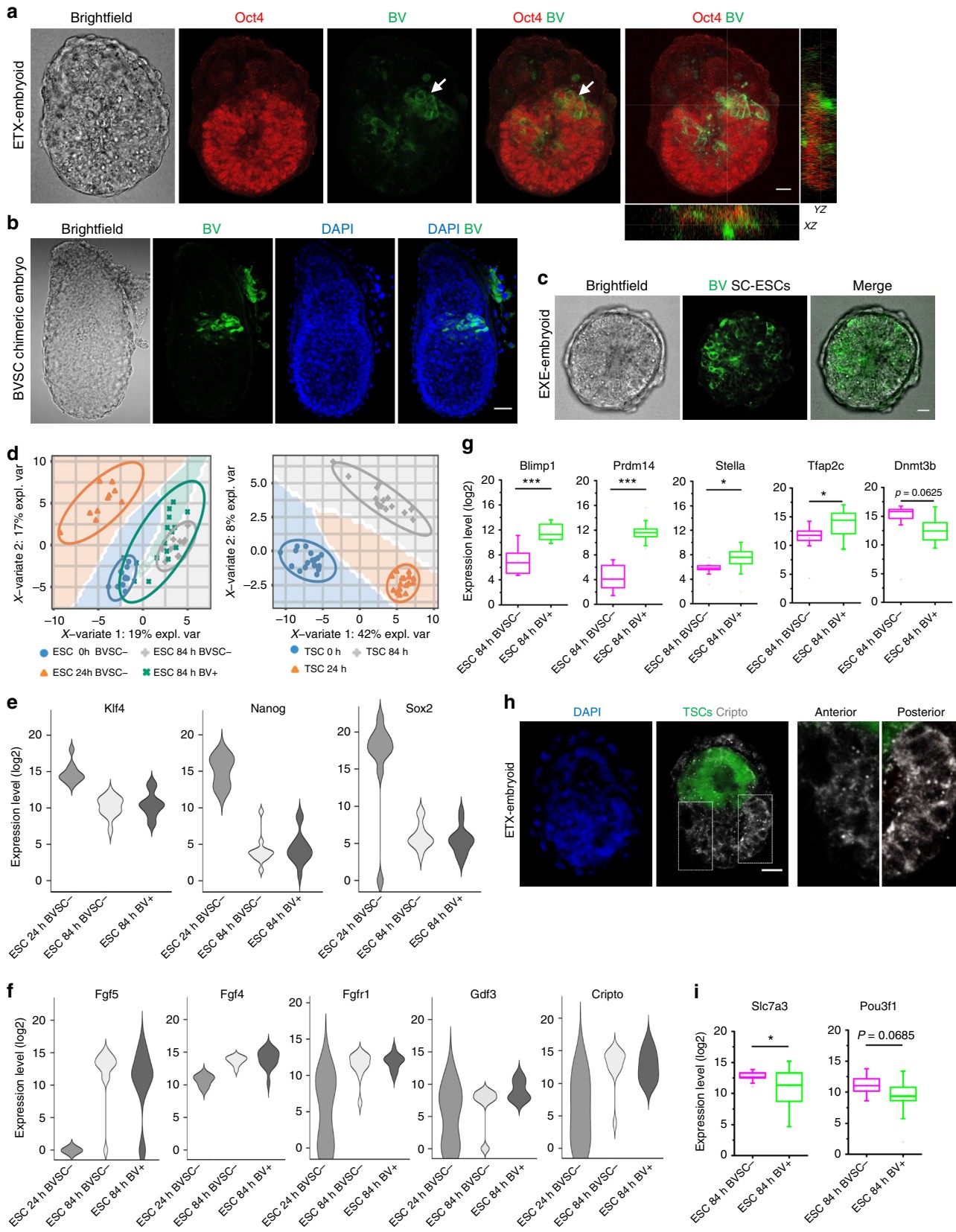

compartment is required for development of the TSC compartment in ETX-embryoids.

**Regionalization of VE during ETX-embryogenesis.** Before mouse embryos initiate gastrulation, asymmetric patterns of gene expression are seen in the VE. This directs the migration of cells from the DVE to form the AVE at E6.5[36,37]. Signaling molecules from the AVE, such as Lefty1 and Cer1, inhibit Nodal signaling in the underlying EPI, leading to asymmetric patterns of gene expression and formation of the anteroposterior axis[2,38,39].

**Fig. 6** Regionalized putative mesoderm formation and PGC precursor-like cell specification in ETX-embryoids. **a** Asymmetrical gene expression of *Blimp*1-mVenus in ETX-embryoids, Oct4 (red) indicates ESC-derived tissues, and the arrow indicates *Blimp*1-mVenus-positive tissues (green), orthogonal xz and yz views are shown. *n* = 6 ETX-embryoids, two experiments. Scale bar, 20 μm. **b** Representative E6.5 chimeric embryo with BVSC ESC-derived tissues. The arrows indicate the localization of *Blimp*1-mVenus-positive cells (green). *n* = 15 embryos, two experiments. Scale bar, 50 μm. **c** Localization of *Blimp*1-mVenus-positive tissues in EXE-embryoids. *n* = 20 EXE-embryoids. Three experiments. Scale bar, 20 μm. **d** Colors and shapes are assigned to the ESC and TSC single-cell groups isolated at 0, 24, and 84 h from ETX-embryoids. The x-axis and y-axis standed for the explained variance in a PLS-DA plot. ESC: *n* = 11 (0 h), 10 (24 h), 11 (84 h, BV-), and 16 (84 h, BV +). TSC: *n* = 18 (0 h), 16 (24 h), 15 (84 h). **e, f** Violin plots of expression levels of naive ESC marker genes (*Klf4*, *Nanog*, *Sox2*) and EPI-specific marker genes (*Fgf5*, *Fgf4*, *Fgfr1*, *Gdf3*, *Cripto*) in BVSC-ESCs, respectively, which reveal that the pluripotency of ESCs decreased as the ESCs differentiation. A background of Ct = 26 was used to obtain expression levels. **g** Single-cell RT-qPCR analysis of the expression of PGC markers (*Blimp1*, *Prdm14*, *Stella*, *Tfap2c*, and *Dnmt3b*) in *Blimp*1-mVenus-positive and *Blimp*1-mVenus-negative cells from the ESC compartments of the ETX-embryoids. Box-plots are generated as described in Fig. 1h. Two-tailed Student's *t*-test, *n* = 11 BV- and 16 BV + single cells (84 h). *P < 0.05, ***P < 0.001. **h** Immunostaining indicates asymmetrical expression of Cripto (gray) in ESC compartment of ETX-embryoids, White boxes, magnified regions showing Cripto negative (left, anterior) and positive (right, posterior) cells. *n* = 10 ETX-embryoids, two experiments. Scale bar, 20 μm. **i** Single-cell RT-qPCR analysis of the expression of the genes (*Slc7a3*, *Pou3f1*) in *Blimp*1-mVenus-positive and *Blimp*1-mVenus-negative cells from the ESC compartments of the ETX-embryoids. Box-plots are generated as described in Fig. 1h. Two-tailed Student's *t*-test, *n* = 11 BV- and 16 BV + single cells (84 h). *P < 0.05

To determine whether XENC-derived tissue in ETX-embryoids mimics DVE/AVE formation in wild-type embryos, we used XENCs that express a *Lefty1*-mCherry reporter to monitor Lefty1 expression. A small number of cells on one side of the XENC-derived layer expressed *Lefty1*-mCherry in 23% ETX-embryoids under culture from 72 to 84 h (Fig. 5a–e and Supplementary Fig. 8a–c). This was confirmed by quantitative assessment of *Lefty1*-mCherry positive cell number in ETX-embryoids (Fig. 5f). We detected that the mCherry positive cells in DVE-like tissue of the ETX-embryoids at 72 h migrated to the anterior side of the embryoids after further culture for 12 h in a ratio of 53% (Supplementary Fig. 8d). This is similar to the distribution of DVE/AVE in wild-type embryos (E5.5-6.5). In EXE-embryoids, however, the distribution of *Lefty1*-mCherry positive cells in XENC-derived tissue was random (Fig. 5h and Supplementary Fig. 8e). We further confirmed that the mCherry positive cells were DVE/AVE-like tissue cells by measuring the expression of specific lineage markers via single-cell PCR. Partial least square-discriminate analysis (PLS-DA)[40] and box-plots revealed that DVE/AVE markers such as *Foxa1*, *Cer1*, *Otx2* and *Sfrp5* were upregulated in the mCherry positive XENC-derived cells (Fig. 5g, i). Markers of posterior VE (*Bmp2*, *Fgf5*) were upregulated in the mCherry negative XEN cells at 84 h compared with mCherry negative cells at 24 h (Fig. 5j). These results suggest that regionalized DVE/AVE-like tissue formed in ETX-embryoids.

**Asymmetric expression of genes specific for mesoderm and PGC precursors.** A crucial event in mouse embryogenesis is the breaking of symmetry in gene expression. This plays a major role in embryonic patterning, resulting in the formation of regionalized mesoderm and the specification of PGCs in the EPI[36,41].

We next sought to determine whether mesoderm induction and PGC specification occurred in ETX-embryoids. Blimp1 (also known as Prdm1) is the key regulator of PGC specification and is expressed in PGC precursors as they emerge from the most proximal posterior portion of the EPI, as well as in the VE[41–43]. *Blimp1*-positive cells also initially express many genes involved in embryonic development, particularly mesoderm induction[44]. To identify cells resembling mesoderm cells and PGC precursors, we used a *Blimp1*-mVenus and *Stella*-ECFP (BVSC) double-transgenic ESC line[45,46] to generate ETX-embryoids. Notably, in contrast to EXE-embryoids, ETX-embryoids showed asymmetric expression of Blimp1-mVenus, which was confined to a domain in the ESC compartments extending from the boundary of the ESC and TSC compartments at 84 h, but ECFP was not detected (Fig. 6a, c and Supplementary Fig. 9a–c). This phenomenon resembled the events in chimeric embryos at E6.5 (Fig. 6b and Supplementary Fig. 9d) after injection of BVSC-ESCs

into embryos at the 8-cell stage[47], as well patterns seen in previous reports[46]. We did not detect *Stella*-ECFP PGCs in BVSC chimeric embryos before E7.5, consistent with the absence of *Stella*-GFP observed in embryos between E5.5-E7.5, which has been reported previously[48].

To examine changes in gene expression patterns in different compartments at different stages, we analyzed the gene expression levels in single cells from the three compartments at 0, 24, and 84 h using single-cell qPCR. PLS-DA components suggested that the three types of stem cells differentiated along with the development of the ETX-embryoids (Fig. 6d). Exit from the naive pluripotent state is required to generate the pro-amniotic cavity in mouse embryos[49]. Therefore, we analyzed patterns of gene expression for cells from the ESC compartment of ETX-embryoids at 24 and 84 h. Naive ESC markers, such as *Nanog*, *Klf4*, and *Sox2*, were downregulated (Fig. 6e) and EPI markers, such as *Fgf5*, *Fgf4*, *Fgfr1*, *Gdf3*, and *Cripto*, were upregulated (Fig. 6f), consistent with the gene expression patterns in the EPI cells of wild-type embryos during gastrulation[49].

Next, we compared the gene expression patterns of *Blimp1*-mVenus-positive (BV[+]) and *Blimp1*-mVenus-negative (BV[−]) cells in ETX-embryoids at 84 h. As expected, PGC markers, such as *Blimp1*, *Prdm14*, *Stella* and *Tfap2c*, were higher in mVenus-positive cells, whereas there was a trend toward *Dnmt3b* downregulation (Fig. 6g). Markers known to be elevated in the region opposite to the mesoderm in E6.5 embryos, such as *Pou3f1* and *Slc7a3*, tended to be upregulated in mVenus-negative cells isolated from the region opposite to that of BV[+] cells (Fig. 6i). BV[+] cells also initially express mesoderm genes[44]. Interestingly, we found the mesoderm markers, such as *Flk1*, *Hhex*, and *Hand1*, were upregulated in BV[+] cells (Supplementary Fig. 9f). We further confirmed mesoderm formation by immunostaining for Cripto, a mesoderm marker, which asymmetric expressed at the posterior side in the ESC compartment (Fig. 6h and Supplementary Fig. 9e). Single-cell qPCR revealed that anterior EPI markers (*Fgf4*, *Mixl1*) were upregulated in 84 h BV[+] cells compared to 84 h BV[−] cells (Supplementary Fig. 9g). These results indicate that the formation of regionalized mesoderm and specification of PGC precursor-like cells occurred in ETX-embryoids.

**ETX-embryoids initiate implantation.** We further analyzed the expression of genes related to blastocyst activation and embryo implantation[50,51]. Markers known to be elevated during implantation and post-implantation, such as *Alk2*, *Erk2*, *Gata3*, *Cb1*, *Cb2*, *ErBb1*, *ErBb4*, and *Igf2*, were upregulated in cells isolated from TSC compartment in ETX-embryoids compared to those of TSCs (Supplementary Fig. 10a, b), suggesting that they

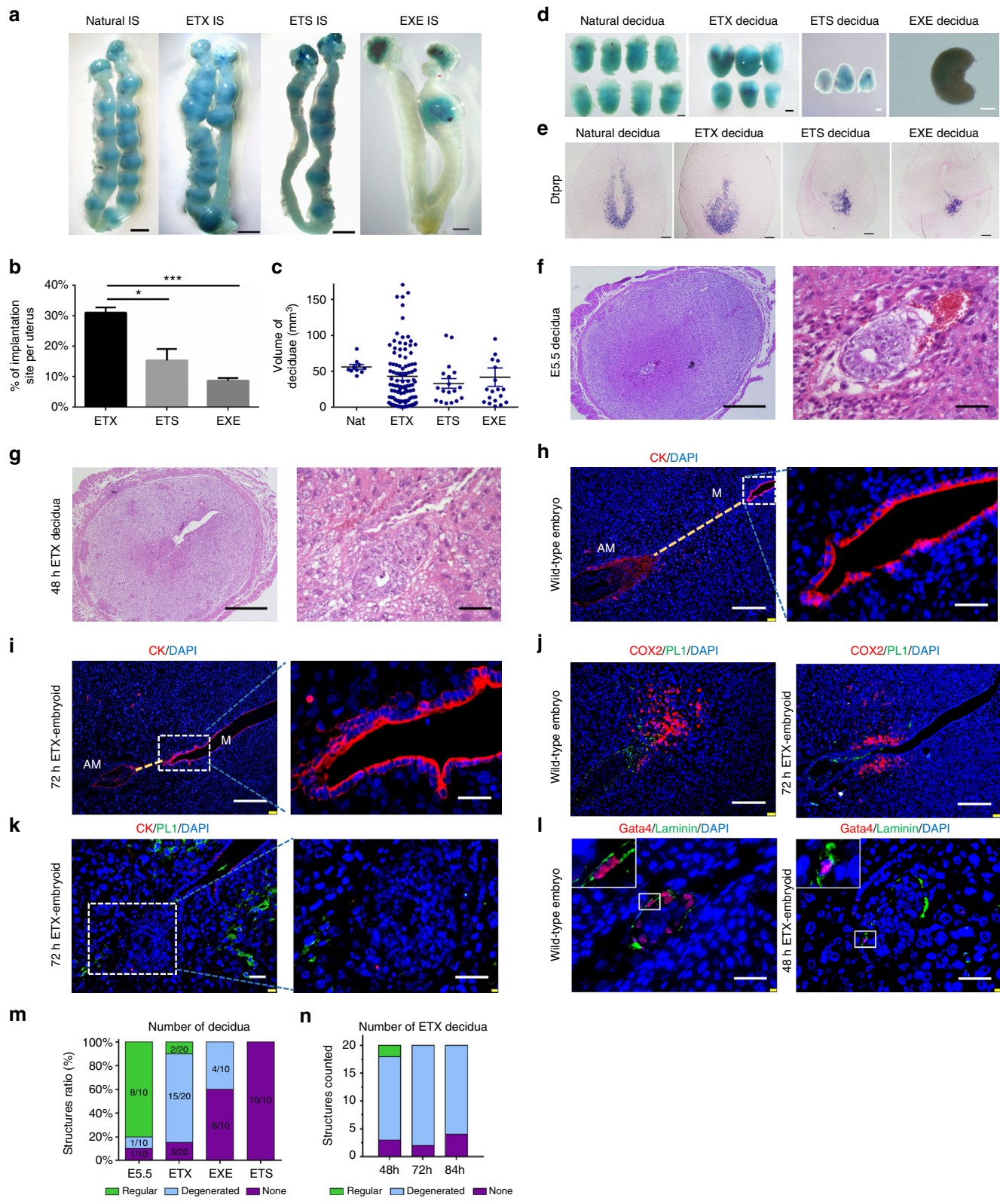

may be able to initiate implantation after transfer into the uterus of a pseudopregnant female.

To test this possibility, we transferred 36 h ETX-embryoids into the uteri of 2.5 and 3.5 dpc pseudopregnant mice. We observed a higher percentage of implantation sites when using 3.5 dpc mice, so this method was used for further study. The highest

percentage of implantation sites was observed for ETX-embryoids compared with ETS- and EXE-embryoids (Fig. 7a, b, d and Supplementary Fig. 12a). The average size of decidual tissues for the three types of reconstructed embryos was comparable, although the individual sizes were variable (Fig. 7c). The identity of these decidual tissues was confirmed via in situ hybridization

**Fig. 7** Implantation initiation of the ETX-embryoids. **a** Representative implantation sites (IS) in uteri visualized via blue dye after transfer of wild-type embryos and different embryoids. Scale bar, 2 mm. **b** Percentages of IS per uterus transferred with different embryoids. Two-tailed Student's $t$-test, $n = 5$ mice per group and each recipient transferred with 20 embryoids, three experiments. *$P = 0.0193$, ***$P = 0.0003$. **c** Deciduae volume of E6.5 embryos and different embryoids after transplantation. $n = 10$ wild-type embryos, 131 ETX-embryoids, 18 ETS-embryoids, 17 EXE-embryoids. **d** Representative deciduae of wild-type embryos and different embryoids. Scale bar: black, 500 μm; white, 200 μm. **e** Detection of *Dtprp* in deciduae of wild-type embryos and different embryoids via in situ hybridization. $n = 5$ for each group. Scale bar, 500 μm. **f, g** Representative cross-section by H.E. staining of E5.5 embryo and ETX-embryoid deciduae. $n = 4$ wild-type embryos and 6 ETX-embryoids. Scale bar, left panel, 500 μm, right panel, 50 μm. **h–j** Immunostaining of E6.5 embryos and ETX-embryoids' IS. **h, i** CK (red). The yellow dotted line shows the uterine axis M-AM. M, mesometrial pole; AM, antimesometrial pole. Scale bar, left panel, 500 μm, right panel, 50 μm. **j** COX2 (red) and PL1 (green). Bar, 500 μm. $n = 5$ wild-type embryos, 5 ETX-embryoids. **k** Immunostaining of CK (red) and PL1 (green) in ETX-embryoids' IS. Differentiated PL1 + TGCs directly connect with stromal cells of the uterus. White boxes show PL1 and CK-positive cells. Bar, 50 μm. Bar in zoomed, 10 μm. **l** Immunostaining of Gata4 (red) and Laminin (green) in E5.5 embryos from **f** and ETX-embryoids' IS from **g**. $n = 5$ wild-type embryos, three ETX-embryoids. Scale bar, 50 μm. **m** Ratios of embryos with different morphologies detected in the deciduae deduced by E5.5, ETX-, EXE-, and ETS-embryoids. $n = 10$ deciduae in wild-type, EXE, and ETS-embryoids, 20 in ETX-embryoids. **n** Morphological analysis of ETX-embryoid in deciduae. "Regular" indicates the ETX-embryoid morphology similar to the wild-type embryos. "Degenerated" indicates the ETX-embryoid degenerated. "None" indicates ETX-embryoid components wasn't detected. $n = 20$ deciduaes for each stage. Error bar are means ± s.e.m. Source Data of **b**, **c**, **m**, **n** are provided as the Source Data file

for *Dtprp* (Fig. 7e). Coordinated uterine-embryonic axis formation and decidual remodeling are hallmarks of mammalian post-implantation embryo development[52]. Haemotoxylin and eosin (H.E.) staining indicated that ETX-embryoids established the uterine-embryonic axis (Fig. 7f, g, h, i and Supplementary Fig. 11a–d). We further examined the expression of implantation markers at the implantation sites, such as Cyclooxygenase-2 (COX2)[53] and Placental Lactogen-I (PL1)[54,55], which marks trophoblast giant cells, as well as Cytokeratin (CK) in the luminal epithelium (LE) (Fig. 7i, j). The expression patterns of these genes at the implantation sites were similar to those seen in E6.5 wild-type embryos (Fig. 7h, j). In addition, the results of CK and PL1 co-immunostaining showed that CK had no signal around the ETX-embryoid at 72 h (Fig. 7k), combined with previous H.E. staining results (Fig. 7g), which suggested clearance of uterine luminal epithelial cells[56] (Supplementary Fig. 11i). These results indicated that the ETX-embryoids recapitulate key events of uterine implantation, such as discrete decidualization, vascular anastomosis, patterned *Dtprp* expression in the decidua, and clearance of uterine luminal epithelium.

For transferred ETX-embryoids, E5.5 embryo-like structures were detected in the decidual tissues at 48 h after transplantation through H.E. staining (Fig. 7f, g). These structures were surrounded by a basement membrane of Laminins (Fig. 7l), which is crucial for early developmental events[32]. No such structures were seen for ETS- or EXE-embryoids (Fig. 7m). However, at later time points (72 and 84 h), we failed to detect ETX-embryoids with a normal developmental morphology (Fig. 7n and Supplementary Fig. 11c, d), suggesting that ETX-embryoids were degraded after implantation initiation. This did not result from immune rejection, as similar results were seen using NOG immune-deficient mice as the recipients (Supplementary Fig. 11e, f). We characterized the cytoarchitecture of these ETX-embryoids by isolating them from deciduae 3 days after transplantation and subjecting them to histochemical and PCR analyses. A total of 17 structures were detected from nine recipient mice. Many identifiable tissues could be identified in ETX-embryoids, but most ETX-embryoids had structural defects (Supplementary Fig. 12b–g). Immunostaining for Gata4 and Laminin confirmed that the implantation sites contained XENC-derived tissues that secreted Laminins (Supplementary Fig. 12h). Comparing ETX-embryoids and wild-type embryos labeled for Laminins and Gata4 (Supplementary Fig. 11g, h), we found that implanted ETX-embryoids lack a structure similar to the outer layer surrounding the entire conceptus, which forms a safe environment for the developing embryo. Altogether, these results

indicate that ETX-embryoids can initiate implantation response after transplantation into a pseudopregnant mouse, although they exhibited limited developmental potential.

## Discussion

In this study, we found that the three types of blastocyst-derived stem cells have a remarkable ability to self-assemble into a structure (ETX-embryoids) that has embryonic architectural features when cultured under nonadherent-suspension-shaking conditions (Fig. 8). Resulting embryo-like structures represent a powerful model system for studying embryogenesis and post-implantation morphogenesis. ETX-embryoids undergo a series of molecular and morphogenic transitions similar to those seen for wild-type embryos, such as lumenogenesis, the formation of DVE/AVE-like tissues, and the induction of mesoderm and PGC precursors. We also demonstrate that XENC-derived tissues play critical roles in polarization and lumenogenesis events in EXE- and ETX-embryoids. In addition, the length of time from initiation of ETX-embryoid assembly to PGC specification (when Blimp1 expressed) is close to that observed for wild-type embryos (E4.5 to E6.5). This suggests that the ETX-embryoid assembly system closely mimics embryogenesis in vitro, enabling exploration of the molecular mechanisms driving these events. Using this system, we plan to investigate the signaling pathways mediating both homo- and heterotypic interactions between these stem cell types that are required for this spatially ordered assembly, as well as the pathways underlying interactions between the different stem cell-derived tissues.

EBs or micro-patterned colonies derived from ESCs have previously been shown to polarize and to carry out lumenogenesis, but they do not execute many of the spatial and temporal events observed in embryogenesis due to the lack of signals from extra-embryonic tissues[17–19]. Combining ESCs and TSCs in a 3D ECM-scaffold can generate ETS-embryoids whose morphogenesis is similar to that of wild-type embryos, including the induction of asymmetric patterns of gene expression for markers of mesoderm and PGC without signals from the DVE or AVE[9]. However, we have shown that ETS-embryoids, when cultured in simpler culture conditions (see the Method section), do not recapitulate embryogenesis. Our EXE-embryoids formed rosette-like structures and cavities, but did not break symmetry, whereas ETX-embryoids showed events similar to early post-implantation development, even the formation of DVE and AVE-like tissues. Therefore, our studies indicated that XENC-derived tissues play important roles in embryogenesis, and interactions among the three compartments are required for the in vitro development

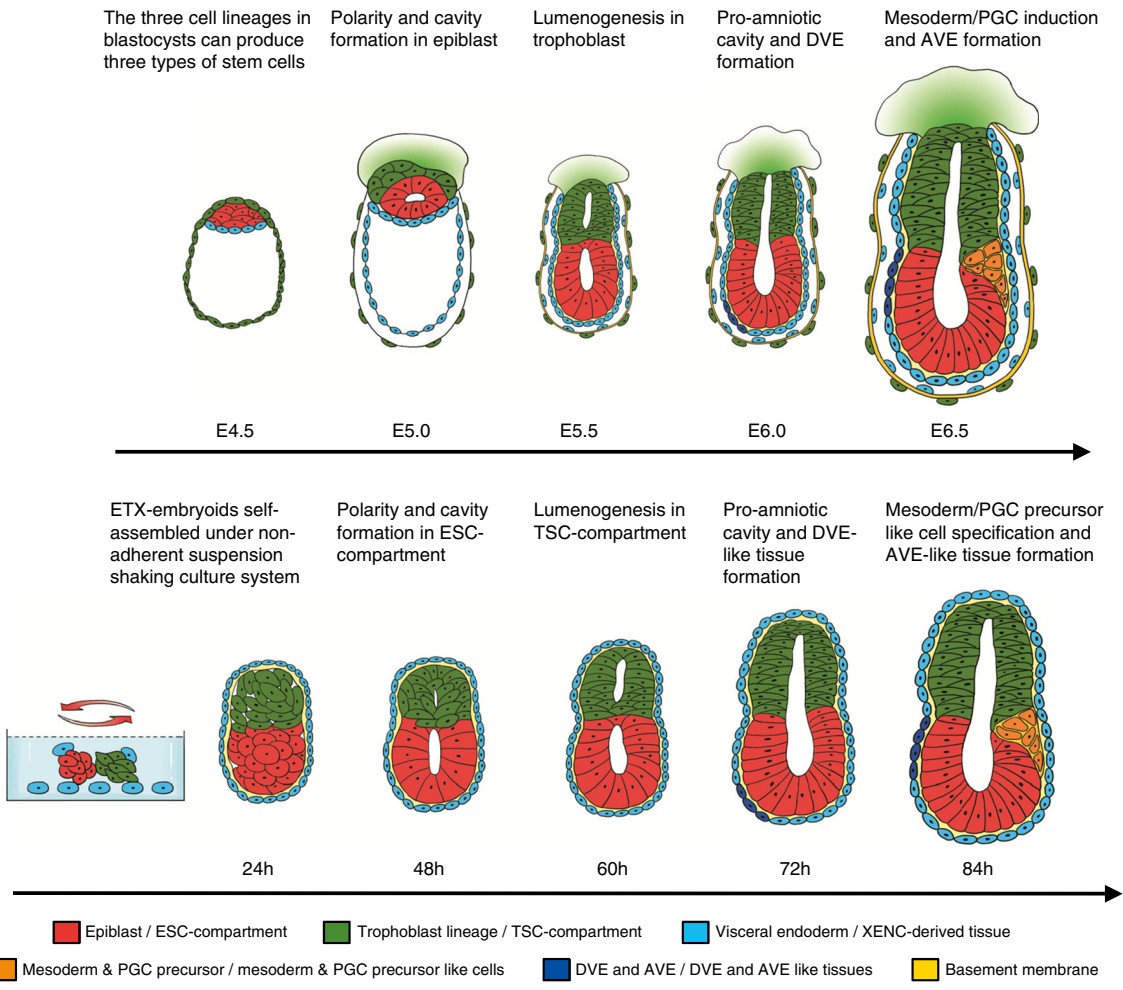

**Fig. 8** Comparison of the developmental characteristics of a wild-type conceptus to that of ETX-embryos. The ETX-embryos, with spatial composition resembling that of post-implantation embryos, can be self-assembled by aggregation of the three types of blastocyst-derived stem cells under nonadherent-suspension-shaking culture condition. When allowed to develop in vitro, the ETX-embryos can be seen as a model to mimic the events, such as polarity, lumenogenesis, DVE and AVE formation, mesoderm and PGC induction, which occur in wild-type embryos. However, the ETX-embryos lack the outer layer of the trophoblast and parietal endoderm together with the basement membrane between them, which surround the entire conceptus and protect their development in vivo

and patterning of self-assembled embryos. Further investigations are needed to better understand the mechanisms governing interactions among the three compartments, and how XENC-derived tissues contribute to development.

ETX-embryoids successfully mimicked early events of natural embryogenesis, even initiating implantation to trigger decidual tissues with typical local vascular permeability and breach of uterine luminal epithelium. However, ETX-embryoids exhibited limited developmental potential after implantation initiation. This could be because ETX-embryoids lack structures such as the parietal yolk sac and Reichert's membrane that function to protect the embryo and to passively filter nutrients[3,57,58]. Normally during gastrulation, the trophoblast-derived giant cell layer and the parietal endoderm, together with the Reichert's membrane, surround the entire conceptus[5,54]. Methods to further improve the in vitro assembly system so as to add more complex structures such as these are clear targets of future studies.

## Methods

**Mice**. All animal procedures and all of the mouse work were approved by the Animal Care and Use Committee of China Agriculture University (Permit Number: SKLAB-2016-01-04). CD1 and 129 mice were obtained from Beijing Vital River Laboratory Animal Technology Co., Ltd. Actin-GFP mice were obtained from Shaorong Gao's Laboratory in Tongji University. All mice were maintained in specific pathogen-free (SPF) conditions with a 12-h dark / 12-h light cycle between 06:00 and 18:00 in a temperature controlled room (22 ± 2 °C) with free access to water and food.

**Stem cell lines and cell culture**. G4 and G4-ACTB-DsRed-MST ESCs were obtained from Dr. Andras Nagy, Kristina Vintersten and Marina Gertsenstein's Laboratory in Mount Sinai Hospital & the Samuel Lunemfeld Research Institute. *Blimp1*-mVenus and *Stella*-ECFP (BVSC) double-transgenic ESC line, expressing membrane-targeted Venus (mVenus) under the control of *Prdm1* (*Blimp1*) regulatory elements and enhanced CFP (ECFP) under the control of *Dppa3* (*Stella/Pgc7*), were obtained from Dr. Mitinori Saitou[45,46].

For generation of *Lefty1*-mCherry mouse XEN cell line, the *Lefty1* promoter and fluorescent protein were used to integrated into XEN cells by the Sleeping Beauty transposon system, which consists of two components: the transposon vector modified from pT2/LTR7-GFP (addgene#62541) and the transposase vector (pCMV(CAT)T7-SB100, addgene #34879). PT2/LTR7-GFP was added with "puro" castle, which could express puromycin N-acetyltransferase and GFP CDS region was replaced by mCherry. DNA fragment, which was amplified from pT2/LTR7-GFP by primer PT2-F (GTTTGGACAAACCACAACTAGAATG) & R (TCTAAAGCCATGACATCATTTTCTG), coupled with "puro" castle and mCherry CDS region to acquire the vector named PT2-Puro-mCherry using NEBuilder®HiFi DNA Assembly Master Mix (NEW ENGLAND BioLabs). The promoter sequence of the *Lefty1* gene was amplified from the DNA of mouse ES cells using *Lefty1*-F (GTCCGGTGGGGAATCACATT) & R (AAAGGGTCTTGAGTCTGCGG). Then the promoter sequence of the *Lefty1* gene was added to PT2-Puro-mCherry by NEBuilder®HiFi DNA Assembly Master

Mix (NEW ENGLAND BioLabs), in which PT2-Puro-mCherry was linearized by *Eco*RI and the overlap sequence containing *Eco*RI sequence was added to the promoter sequence of the *Lefty1* gene. The vector was named PT2-Puro-mCherry-Lefty1. Then PT2-Puro-mCherry-Lefty1 and pCMV(CAT)T7-SB100 were transfected into mouse XEN cells and screened by 2 μg per mL puromycin for subsequent analyses.

For generation of *Lamc1* (laminin, gamma 1) knockout cell lines, two sgRNA sequence targeting (GGAGTACTGCGTGCAGACTGGGG and GCTTTGCCACCAGGTGGTGTCGG) *Lamc1*'s exon1 were used for constructing a dual sgRNA vector. Fragments containing two sgRNA sequences amplified from PUC-U6-sgRNA-Kana (from Huang Xingxu lab) using primers (*Lamc1*sg1-F: atgcgtctcacaccggagtactgcgtgcagactggtttttagagctagaaatagcaag and *Lamc1*sg2-R: atgcgtctcgaaacacaccacctggtggcaaagcggtgtttcgtcctttccacaag). The PCR product was digested with *Bsm*BI and ligated with linearized PX330-GFP. The Vectors has been transfected into ESCs and XENCs using Lipofectamine 3000 (L3000001, Thermo Fisher Scientific), respectively. Then GFP-positive cells were sorted using flow cytometry and diluted for monoclonal growth. DNA was extracted from the obtained monoclonal cells and subjected to PCR amplification and sequencing. Knockout-positive cells were selected for follow-up experiments.

For generation of *Nodal* knockout cell lines, gRNA sequence targeting (GCCCTCGTCACCGTCCCCTCTGG) *Nodal*'s exon1 were used for constructing sgRNA vector. Primers for *Nodal* sgRNA (*Nodal*-sgF: caccgccctcgtcaccgtcccctc and *Nodal*-sgR: aaacgaggggacggtgacgagggc) were annealed and the annealed product was ligated to linearized PX330-GFP by *Bbs*I. The Vectors has been transfected into ESCs using Lipofectamine 3000 and GFP-positive cells were sorted using flow cytometry and diluted for monoclonal growth. DNA was extracted from the obtained monoclonal cells and subjected to PCR amplification and sequencing. Knockout-positive cells were selected for follow-up experiments.

For generation of *p53* knockout ESC line, the sgRNA sequence for *p53* mutation was cloned into the PX330-GFP vector (addgene, #48138) using CRISPR-Cas9 system[59]. The vector was transfected into G4 ESCs by Lipofectamine 3000 (L3000001, Thermo Fisher Scientific). Three days after transfection, GFP-positive cells were sorted and plated on 6-well plate. Colonies were picked, and the knockout regions were amplified by PCR using primers (F: CATCCAGGCGGGAAATAGAGAC; R: CCTGACTGTGTGTAAACTAGGC) and sequenced. By aligning with the wild-type sequence, homozygous knockout cell colonies were identified and selected for further study.

ESCs were cultured in Dulbecco's Modified Eagle Medium (DMEM) (11960069, Thermo Fisher Scientific) with 15% (v/v) FBS (Gibco, ES cell qualified), 2 mM GlutaMAX (35050061, Thermo Fisher Scientific), 0.1 mM β-mercaptoethanol (21985-023, Thermo Fisher Scientific), 0.1 mM MEM non-essential amino acids (11140050, Thermo Fisher Scientific), 1 mM sodium pyruvate (11360070, Thermo Fisher Scientific), 1% penicillin–streptomycin (15140122, Thermo Fisher Scientific), 1000 IU LIF (130-099-895, Miltenyi Biotec), 1 μM PD0325901 (4423, Tocris), and 3 μM CHIR99021 (4192, Tocris). ESCs were cultured at 37 °C and 5% $CO_2$ on gelatinized and feeder-covered tissue culture grade well plates, and passaged when 80% confluency is achieved.

TSCs were derived from E3.5 blastocysts flushed from the uterus of 5-weeks-old female CD1 mouse[15,60]. The blastocysts were cultured on mitotically inactivated mouse MEF cells (feeder cells) with TSC culture medium: advanced RPMI 1640 (11875-093, Thermo Fisher Scientific) supplemented with 20% (v/v) FBS (Gibco, ES cell qualified), 2 mM GlutaMAX (35050061, Thermo Fisher Scientific), 0.1 mM β-mercaptoethanol (21985-023, Thermo Fisher Scientific), 1 mM sodium pyruvate (11360070, Thermo Fisher Scientific), 1% (v/v) penicillin–streptomycin (15140122, Thermo Fisher Scientific), 25 ng per mL recombinant human FGF4 (235-F4-025, R&D) and 1 μg per mL heparin (07980, Stem Cell). Until the outgrowth formed on day 4, they were subsequently disaggregated by incubated in 0.1% trypsin-EDTA for 5 min in the incubator. Add fresh mouse 70% embryonic fibroblast-conditioned medium containing FGF4 and heparin to stop the reaction and return to the incubator. Replace the medium every other day until TS cell colonies can be observed, and change medium with TSC culture medium to maintain the cells. TSC expressing EGFP (TSC-EGFP) cell lines were derived from wild-type TSCs which were transfected with the PB-UBC-EGFP vector by JetPrime (114-15, Polyplus transfection). TSCs were cultured at 37°C and 5% $CO_2$ on gelatinized and feeder-covered tissue culture grade well plate and passaged when 80% confluency is achieved. Culture medium was changed daily.

XENCs were derived from E3.5 blastocysts flushed from the uterus of 5-weeks-old female 129 mouse using modified XENC conditions[61]. Mouse blastocysts were cultured on feeder cells until they formed an outgrowth on day 4 that was subsequently disaggregated by incubated in 0.1% trypsin-EDTA for 5 min in the incubator. Add fresh mouse ES cell medium without PD0325901 and CHIR99021 to stop the reaction, and replace the medium every other day until XEN cell colonies can be observed. XEN cell colonies was maintained with standard XEN medium: advanced RPMI 1640 (11875-093, Thermo Fisher Scientific) supplemented with 15% (v/v) FBS (Gibco) and 0.1 mM β-mercaptoethanol (21985-023, Thermo Fisher Scientific), 1% penicillin–streptomycin (15140122, Thermo Fisher Scientific). XENC-EGFP cell line was derived directly from Actin-EGFP mouse blastocyst. For feeder-free culture, XENCs were maintained on tissue culture grade well plate coated with gelatin. XEN cells were cultured at 37 °C and 5% $CO_2$. Passaged once they reached 90% confluency. Culture medium was changed daily.

**Generation of self-assembled embryos.** ESCs, which reached 80% confluency, were washed once with PBS (14190-094, Gibco) and were incubated for 5 min at 37 °C in TrypLE (12605010, Thermo Fisher Scientific). TSCs and XENCs were dissociated to single cells by incubation with 0.1% trypsin-EDTA (25300054, Thermo Fisher Scientific) at 37 °C. Cells were pelleted by centrifugation at 1000 rpm for 5 min. The pellet was re-suspended in reconstructed embryo medium by pipetting to single cells. The cells were counted automatically with automated cell counter.

For ETX-embryoids, mix ESCs, TSCs, and XNECs at cell number of $1 \times 10^5$, $1 \times 10^5$, and $2 \times 10^5$, respectively (for ETS-embryoids, mix ESCs and TSCs at cell number of $1 \times 10^5$ and $1 \times 10^5$, respectively; for EXE-embryoids, mix ESCs, and XENCs at cell number of $1 \times 10^5$ and $1 \times 10^5$, respectively) with 2 mL of reconstructed embryo medium per 35 mm non-treated cell culture dish. Move the dish on the horizontal rotators at 37 °C, 5% $CO_2$, and 100% humidity. The rotation rate is at 60 rpm per min. The reconstructed embryo medium includes 39% advanced RPMI 1640 (11875-093, Thermo Fisher Scientific) and 39% DMEM (11960069, Thermo Fisher Scientific) supplement with 17.5% FBS (Gibco, ES cell qualified), 2 mM GlutaMAX (35050061, Thermo Fisher Scientific), 0.1 mM β-mercaptoethanol (21985-023, Thermo Fisher Scientific), 0.1 mM MEM non-essential amino acids (11140050, Thermo Fisher Scientific), 1 mM sodium pyruvate (11360070, Thermo Fisher Scientific), 1% penicillin–streptomycin (15140122, Thermo Fisher Scientific). Changed medium with 2 mL fresh reconstructed embryo medium every day.

EXE-embryoids formed consistently and efficiently, the functional experiments can be performed without selection of regular or irregular structures. For the functional experiments of ETX-embryoids and ETS-embryoids, we selected the regular structures based on their morphology, and the appropriate proportions of ESC and TSC compartments by indication of TSC-EGFP under stereo fluorescence microscope.

**Wild-type embryo recovery and in vitro culture.** Blastocysts were recovered from E4.5 superovulated pregnant female mice by uterine flushing with M2 (MR-015-D, Millipore). Culturing embryos in IVC medium after removal of the mural trophectoderm in vitro on 8-well μ-Slides (80826, Ibidi)[62]. For recovery of E5.5, E5.75, and E6.5 embryos, dissection of the decidua from the uterus describe as the reported protocol[63]. For EPI-explants from CD1 mice were dissected on day E5.0 as previous reported[28] and cultured with ETX-embryoids medium in vitro.

**Immunofluorescence staining.** Mouse embryos and reconstructed embryos were fixed with 4% paraformaldehyde (DingGuo, AR-0211) at 4 °C for 20 min and washed three times with a wash buffer (PBS containing 0.1% Tween-20). Then, the embryos were permeabilized with 0.5% Triton X-100 (H5142, Promega) for 30 min at room temperature and were incubated with primary antibodies in blocking buffer (3% BSA, 0.3% Triton X-100) overnight at 4 °C. After being washed to remove unbound primary antibodies, cells were incubated with secondary antibody in blocking buffer overnight at 4 °C. Cell nucleus was stained with DAPI, followed by twice washes with washing buffer, then were mounted in Fluoroshield Mounting Medium with DAPI (104140, abcam). Images were captured with a Nikon A1 confocal microscope.

**Imaging with processing and analysis.** Images for mouse embryos and reconstructed embryos were acquired using a Nikon A1 confocal microscope with a 20× or 40× objective. All analyses were carried out using open-source image analysis software "Fiji". Tissues' volume and cell numbers for reconstructed embryos and wild-type embryos were estimated and immunofluorescence intensity measurements for PCX distribution analysis was also performed using Fiji software[9].

**High-throughput single-cell qPCR and data processing.** For single-cell isolation, certain cells at different developmental stages (0, 24, and 84 h) were collected from two types of ETX-embryoids, which were reconstructed with *Lefty1*-mCherry XENC (LC-ETX-embryoids) and BVSC ESC (BVSC-ETX-embryoids) reporter cell lines, respectively. To isolate the single cells, we only select the LC-ETX-embryoids with DVE/AVE-like tissues identified by asymmetry expression of the mCherry proteins, and select the BVSC-ETX-embryoids with asymmetry expression of the mVenus proteins in the ESC compartment near the boundary between the ESC- and TSC compartments. For LC-ETX-embryoids, 0, 24, and 84 h LC negative ESC-derived cells (LC⁻ XEN) as well as 84 h positive ESCs (BVSC⁺ ES) from the opposite side of LC⁻ XEN were collected. For BVSC-ETX-embryoids, 0, 24, and 84 h BV negative ESC-derived cells (BVSC⁻ ES) as well as 84 h positive ESCs (BVSC⁺ ES) from the opposite side of BVSC⁻ ES were collected. In addition, single cells from TSC compartments in the ETX-embryoids assembled with BVSC-ESCs at 0, 24, and 84 h were collected based on EGFP reporter.

For single-cell separation, all ETX-embryoids were incubated in 0.1% trypsin-EDTA (Invitrogen, Carlsbad, USA) for 5 min at 37 °C and then transferred into M2 medium (MR-015-D, Millipore). After gentle repeated mouth pipetting for several times, single cells were isolated with a finely pulled glass tip. On average, 30–40

cells were collected for each type. Each cell was washed three times in Dulbecco's phosphate-buffered saline (Invitrogen, Carlsbad, USA) containing 0.1% BSA (Sigma-Aldrich, MO, USA), and placed into reverse transcription-polymerase chain reaction (RT-PCR) master mix for lysis, sequence-specific reverse transcription, and pre-amplification, as described as previous report[64].

For design and validation of primers, genes which regulate cell lineage differentiation (mainly from E3.5 to E6.5 stage) and participate in dominant signaling pathway (e.g., Wnt, Nodal signaling pathway) were selected, according to the embryogenesis of mouse. The mRNA sequences for each interested gene were retrieved from NCBI and only common regions were used for genes with different transcripts. Gene-specific qRT-primers were designed with Primer3 software within the length of 100–250 bp. All primers had been tested using cDNA of mouse embryos (mixture of E5.5 and E6.5) for amplification efficiency and specificity (listed in Supplementary Data 1).

Single-cell sequence-specific pre-amplification similar to previous report[65]. A total of 96 primer sets were pooled to a final concentration of 0.1 μM for each primer. Individual cells isolated from embryos were transferred into sterile microtubes loaded with 5 μL of RT-PCR master mix containing 2.5 μL of CellsDirect reaction mix (Invitrogen, Carlsbad, USA), 0.5 μL of primer pool, 0.1 μL of RT/Taq enzyme (Invitrogen, Carlsbad, USA), and 1.9 μL of nuclease free water (Invitrogen, Carlsbad, USA) in each tube. Tubes were immediately frozen on dry ice. After brief centrifugation at 4 °C, tubes were immediately placed in PCR machine. Cell lyses and sequence-specific reverse transcriptions were performed at 50 °C for 1 h. Afterwards, reverse transcriptase inactivation and Taq polymerase activation were achieved by heating to 95 °C for 3 min. Subsequently, in the same tube, cDNA went through 20 cycles of sequence-specific amplification by denaturing at 95 °C for 15 s, and annealing and elongation at 60 °C for 15 min. To avoid evaporation, the resulting products were stored at −80 °C.

For high-throughput microfluidic single-cell quantitative PCR, all preamplified cDNA products were validated by real-time PCR system (ABI 7500) using endogenous control gene *Actinb*, only threshold crossing (Ct) value within 9 to 12 were selected for subsequent experiment. Amplified single-cell samples were analyzed with SsoFast EvaGreen Supermix with Low ROX (Bio-Rad, California, USA) and individual qPCR primers in 96 × 96 dynamic arrays on a Biomark System (Fluidigm, San Francisco, USA). Ct values calculation were conducted with the BioMark Real-Time PCR Analysis software (Fluidigm, San Francisco, USA).

For Single-Cell Data Processing and Visualization, all raw Ct values obtained from the BioMark System were converted into Log2 relative expression levels by subtracting the values from the assumed background Ct value of 26. Samples with low or absent expression of endogenous control genes (*Actinb* and *Gapdh*) were excluded from subsequent analysis. PLS-DA was applied to the expression data to discriminate cell groups in different ETX-embryoids with R package mixOmics[66]. Hierarchical clustering was performed using Euclidean distances, and dendrograms were displayed along the rowscaled heatmaps using the fluidigmSC package. Box-plots and violin plots were generated with the origin 8.6 and R software.

**BVSC ESC-Chimeric embryo production.** We produced the BVSC ESC-Chimeric embryos by 8-cell embryo injection method by Piezo Micro-manipulation[67]. To ensure the host embryos were injected before compaction, we collected the embryos at two-cell stage and cultured in vitro. To obtain two-cell stage embryos, 5-weeks-old female mice were superovulated by intraperitoneal injection of five international units (IU) of PMSG, followed by 5 IU HCG 46–48 h later, and mated with male mice. Two-cell stage embryos were collected by flushing the oviduct with M2 at E1.5. Embryos were washed in M2 (MR-015-D, Millipore) and then transferred into 10 μL KSOM (MR-020P-5F, Millipore) drops covered with mineral oil on a tissue culture dish. The embryos were maintained at 37 °C with 5% CO2 in an incubator.

About 15 ES cells were injected into 8-cell stage embryos by Piezo Micromanipulation[67,68]. Eight-weeks-old CD1 females mated with vasectomized males were used as pseudopregnant mice. Injected blastocysts were transferred into the uterus of pseudopregnant females at 2.5 days post coitum (dpc). Embryos at the morula stage with injected ESCs were transferred into the oviduct of 0.5 dpc recipients. And we transferred 14-16 embryos per recipient. At E6.5, we collected the chimeric embryos from deciduae as described above.

**Embryo transfer and implantation evaluation.** Eight-weeks-old CD1 females mated with vasectomized males were used as pseudopregnant mice. For reconstructed embryos, 36 h reconstructed spheres (ETX-, ETS-, and EXE-embryoids) were transferred into the uteri horns of pseudopregnant recipient CD1 female mice at 2.5 dpc or 3.5 dpc. Sixteen to twenty embryos were transferred per recipient. Implantation sites on Day 6.5 were identified and counted by intravenous injection of 100 μL 1% Chicago Sky Blue solution. The length and radius of each decidua was measured using image analysis software, and the volume was calculated from these measurements as $V = \pi r^2 l$.

**HE staining and immunostaining of implantation sites.** To examine the development potential of self-assembled embryos, we collected decidua from diverse

stage, including 48, 72, and 84 h after transplantation. These reconstructed (ETX, EXE, and ETS) embryos samples were fixed with 4% paraformaldehyde for overnight at 4 °C, dehydrated with gradient alcohols, transferred into xylene and embedded in Paraffin. We used E5.5, E6.0, and E6.5 wild-type embryos as control. Samples were sliced to 5 μm thickness, stained with hematoxylin and eosin, and observed under a microscope. Deparaffinized rehydrated samples were subjected to heat-induced epitope retrieval in citrate buffer at around 95 °C by a microwave oven for 10 min, following cooled down to room temperature and rinsed twice in PBS with 3 min by twice. Permeabilization was performed with 0.5% Triton X-100 in PBS for 20 min at room temperature. After permeabilizing, specimens were washed in PBS for three times (3 min each). Then samples were blocked for 30 min with 5% bovine serum albumin (BSA) in PBS and incubated with primary antibodies against Rabbit anti-CK (ab9377, Abcam), Rabbit anti-COX2 (ab15191, Abcam), Rabbit anti-Laminin (L9393, Sigma), Goat anti-Gata4 (sc-1237, Santa Cruz), or Mouse anti-PL1(sc-376436, Santa Cruz) at 4 °C overnight. The following day, slices were washed three times in PBS for 5 min and incubated with secondary antibody for 30 min at 37 °C. Secondary antibodies were labeled with Alexa Fluorophore 488 or 594 (Invitrogen). Subsequently, sections were rinsed six times in PBS with 5 min each and mounted with Fluroshield mounting medium with DAPI (ab104140, Abcam). Slides were kept at −20 °C until observed. Images were captured with a Leica fluorescent microscope.

**In situ hybridization.** In situ hybridization with digoxygenin (DIG) was performed on cryosections[52]. Mouse-specific cRNA probes for *Dtprp* were used for hybridization. Sections hybridized with sense probes served as negative controls. Frozen sections (8–10 μm) were adhered to poly-L-lysine coated slides (Citotest Labware Manufacturing Co., Ltd, Jiang Su, China) and stored at −80 °C until used. Sections were placed on a slide warmer (37 °C) for 2 min and then rehydrated in PBS, post-fixed in 4% PFA for 15 min at 4 °C, treated with proteinase K (15 μg per mL for 5 min at room temperature), 0.25% acetylated for 10 min and hybridized with DIG-labeled probes overnight at 65 °C. Two 65 °C post-hybridization washes were carried out followed by two washes in 1 × MABT, and 30 min RNase treatment (20 μg per mL) at 37 °C. Sections were blocked for 1 h, and incubated overnight in block with anti-DIG antibody (1:2500, Sigma-Aldrich). After washing, color was developed using NBT/BCIP until a brown precipitate was visible. Slides were counterstained with nuclear fast red, dehydrated and cleared in xylene, and mounted in cytoseal mounting medium. Photographs were taken promptly before fading of the INT/BCIP precipitate occurred.

**Genotype identification of the transferred self-assembled embryos.** Genomic DNA was extracted from DsRed-ESCs, EGFP-TSCs, and ETX-embryoids isolated from decidua tissues using a TIANamp Micro DNA Kit (DP316, Tiangen) according to the manufacturer's instructions. The diagnostic primer set *EGFP* (5′-AGGACGTCATCAAGGAGTTC-3′, 5′-CAGCCCATAGTCTTCTTCTG-3′) and *DsRed* (5′-GTGCTTCAGCCGCTACCC-3′, 5′-AGTTCACCTTGATGCCGTTCT-3′) were used to identify the region of the cells and the samples by PCR. The thermal profile was 95 °C for 5 min, 35 cycles of 95 °C for 30 s, 60 °C for 30 s, and 72 °C for 20 or 40 s, and a final cycle at 72 °C for 5 min.

**Statistics.** Statistical analyses were processed on GraphPad Prism 7.0 software (GraphPad Software, La Jolla, CA) for Windows (including single-cell Log2 expression data). Data had been checked for normal distribution and equal variances with F-test before each parametric statistical test was performed. Where appropriate, two-tailed Student's t-tests were performed with Welch's correction if variance between groups was not equal. Error bars represent standard error of s.e. m or s.d. as specified. Figure legends indicate the number of independent experiments performed in each analysis. Each experiment had been repeated for reproducibility.

Tissue volume (Figs. 1i and 7c, and Supplementary Fig. 4h), percentage and number calculation (Figs. 2h, 2i, 3k, 4c, 4f, 5e, 5f, 7b, 7m, and 7n; Supplementary Figs. 2d, 3c, 4g, 8d, 11f, 12a and 12g) was determined using Excel. The significance analysis was performed using t-test by GraphPad Prism. Bar graphs were drawn by Excel and GraphPad Prism, and scatterplots were plotted using GraphPad Prism.

**Reporting summary.** Further information on experimental design is available in the Nature Research Reporting Summary linked to this article.

## Data availability
The data supporting the findings of this study are available from the corresponding author on reasonable request. Source data for single-cell quantitative PCR experiments (Figs. 1h, 5g, 5i, 5j, 6d–g and 6i; Supplementary Fig. 9f, 9g, 10a, and 10b) and the gene list with corresponding sequences (Supplementary Fig. 10a) have been provided in Supplementary Data 1. Qualifications of the data (Figs. 1i, 2h, 2i, 3k, 4c, 4f, 5e, and 5f; Supplementary Figs. 2d, 3c, 4g, 4h, 8d, and 12f) and embryo transplantation data (Figs. 7b, 7c, 7m, and 7n; Supplementary Fig. 11f, 12a) have been provided in Source Data.

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

## Acknowledgements
We thank Andras Nagy, Kristina Vintersten, and Marina Gertsenstein G4 and G4-ACTB-DsRed-MST ESCs; Mitinori Saitou for BVSC ESC line; Shaorong Gao for Actin-GFP mice. We thank Ivan Bedzhov (Max Planck Institute for Molecular Biomedicine, Germany) for suggestion of Oct4 immunostaining for mouse embryos. This work was supported by China National Basic Research Program (2016YFA0100202), National Natural Science Foundation of China (31571497, 31601941, and 31772601), Plan 111 (B12008) and Research Programs from the State Key Laboratories for Agrobiotechnology, China Agricultural University (2017SKLAB1-2 and 2018SKLAB6-20).

## Author contributions
J.H. conceived and supervised the project. S.Z. performed most of the experiments of generation and identification of the self-assembled embryos; T.C., H.W. and S.K. performed post-implantation embryonic H.E. staining and immunostaining; D.G., L.Z. analyzed statistical data and drew some graphs; N.C., Q.W., and H.L. performed the single-cell qRT-PCR and its analysis; M.Z., J.X., X.L., X.W., H.M. performed the BVSC ESC-chimeric embryo experiment; B.S., and Y.L. helped with molecular vector construction. J.H., S.C., and S.Z. analyzed the data and J.H. wrote the manuscript with helps of J.C.I.B., R.C.W., and Y.Y.

## Additional information

**Competing interests:** The authors declare no competing interests.

