## [Peer Review File · Nature Communications]

This manuscript has been previously reviewed at another journal that is not operating a transparent peer review scheme. This document only contains reviewer comments and rebuttal letters for versions considered at Nature Communications. Mentions of the other journal have been redacted.

Reviewers' Comments:

Reviewer #1:

Remarks to the Author:

I read the manuscript NCOMMS-18-30038-T. "Implantation of self-assembled embryos generated using three types of blastocyst-derived stem cells". Although some (trivial) experiments requested or problems regarding exaggerated claims have not been not substantially addressed, I acknowledge that the authors have made an effort to improve the quality of manuscript.

Hence I suggest the authors revise the manuscript, toning down where adequate some of the claims without the need for extra experiments:

1) I suggest:

- In the title replace "implantation" perhaps by "implantation response" or "implantation initiation" as mentioned in the text (line 365, 378).
- The ETX-structures described in the manuscript are not "embryos" and that is also misleading. They should be referred as "ETX-embryoids", "ETX-structures" or "ETX-embryo-like structures" throughout the manuscript, as in Line 48 of the abstract
- Line 47: In abstract: the ETX-structures did not "formed decidual tissues", but "triggered the formation of decidual tissue in the uterus"

2) Absence of statistics

In the rebuttal, the authors claimed they added statistical charts;

- "We added a statistical chart showing the number of ETX-embryos with cavitated ESC compartments after 72 hours under control, SB431542 and Nodal-/- ESC culture conditions, please see Fig. 4f as following: ".
- "We cultured these ETX-embryos to 84 h, and calculated the statistics on the ratio of AVE and DVE. We added the evidence of DVE/AVE for developmental progression at 72 h and 84 h in the revised version, please see Extended Data Fig. 8d as following."

I cannot find the "statistics" regarding Figure 4f and Ext Data 8d.

3) Novel data regarding implantation

The novel results regarding implantation and presented in Figure 7k are misleading as the white line-box is misplaced (compare with Figure 7l). This indicates that the staining for Elf5 and Cdx2 that should stain "TE cells" is not in the ETX-structure. I suggest the authors remove Figure 7k. Particularly, if the authors are not able to show Cdx2/Elf5 in the natural embryo for comparison.

I also suggest the text "AM" in Figure 7i in moved upwards/downwards, as it is now obstructing the ETX-structure. Again compare the place of the ETX-structure in Figure 7i (AM) with Figure 7k.

If the authors really want to show evidence that the ETX- structures resemble E5.5 embryo, I would expect they to provide not only staining for TE-derived cells, but also ICM-derived cells (Nanog/Oct4) and compare that nicely with natural embryos. As the evidence is lacking, I don't think the authors should claim so strongly that:

Line 359-360 "structures that resembled E5.5 embryos were detected in the decidual tissues at 48 h after transplantation"

Reviewer #2:

Remarks to the Author:

The revised manuscript by Zhang et al. describes a novel method for mimicking embryogenesis by assembling synthetic embryos from three different types of blastocyst-derived stem cells. This is an exciting approach which should be of interest to developmental biologists and stem cell researchers alike.

Although similar embryo-like structures have been described recently (Harrison et al, 2017), the ETX embryos generated here represent a significant advance due to the inclusion of all three cell lineages of the mouse blastocyst.

In their revision, the authors have addressed the major and minor concerns raised. They have provided additional data and quantifications to support their findings. Importantly, they have performed additional experiments to demonstrate that the emergence of DVE-AVE features progresses in a similar manner to natural embryogenesis and expanded their evidence for the initiation of implantation by ETX embryos. The accuracy of the language in the manuscript has also been improved. Together, these changes bring the paper up to a standard that warrants publication in Nature Communications. Finally, similar work has recently been published elsewhere, but I don't see this as a negative issue. The field will certainly benefit from multiple studies showing robustness and reproducibility of this type of methods.

25 Dec 2018

Re: Nature Communications NCOMMS-18-30038A. "Implantation initiation of self-assembled embryo-like structure generated using three types of mouse blastocyst-derived stem cells".

Point by point response for the Reviewers' comments

REVIEWERS' COMMENTS:

Reviewer #1 (Remarks to the Author):

I read the manuscript NCOMMS-18-30038-T. "Implantation of self-assembled embryos generated using three types of blastocyst-derived stem cells".

Although some (trivial) experiments requested or problems regarding exaggerated claims have not been not substantially addressed, I acknowledge that the authors have made an effort to improve the quality of manuscript.

Hence I suggest the authors revise the manuscript, toning down where adequate some of the claims without the need for extra experiments:

1) I suggest:

- In the title replace "implantation" perhaps by "implantation response" or "implantation initiation" as mentioned in the text (line 365, 378).

Response: Thank you for your suggestion. The words “implantation initiation” are a more accurate description than “implantation”. We have replaced “implantation” by “implantation initiation” accordingly in the title and in the text.

- The ETX-structures described in the manuscript are not “embryos” and that is also misleading. They should be referred as “ETX-embryoids”, “ETX-structures” or “ETX-embryo-like structures” throughout the manuscript, as in Line 48 of the abstract

Response: According to your suggestion, we have substituted all "ETX-embryos" into "ETX-embryoids" in the revised manuscript.

- Line 47: In abstract: the ETX-structures did not “formed decidual tissues”, but “triggered the formation of decidual tissue in the uterus”

Response: Many thanks for your suggestion and apologies for any inadequate descriptions. We have revised our abstract according to your suggestion and editorial requests.

2) Absence of statistics

In the rebuttal, the authors claimed they added statistical charts;

- “We added a statistical chart showing the number of ETX-embryos with cavitated ESC compartments after 72 hours under control, SB431542 and Nodal-/- ESC culture

conditions, please see Fig. 4f as following:".

- "We cultured these ETX-embryos to 84 h, and calculated the statistics on the ratio of AVE and DVE. We added the evidence of DVE/AVE for developmental progression at 72 h and 84 h in the revised version, please see Extended Data Fig. 8d as following."

I cannot find the "statistics" regarding Figure 4f and Ext Data 8d.

Response: Thank you for your comments and apologies for any inadequate descriptions.

Please see the "statistics" regarding Figure 4f and Ext Data 8d (Supplementary Figure 8d in the revised version) in the figure legends and the file of "Source Data".

3) Novel data regarding implantation

The novel results regarding implantation and presented in Figure 7k are misleading as the white line-box is misplaced (compare with Figure 7l). This indicates that the staining for Elf5 and Cdx2 that should stain "TE cells" is not in the ETX-structure. I suggest the authors remove Figure 7k. Particularly, if the authors are not able to show Cdx2/Elf5 in the natural embryo for comparison.

Response: Many thanks for your comments and suggestion. We performed Cdx2/Elf5 immunostaining of wild-type embryos for comparison, which showed that Cdx2/Elf5 localization in ETX-structure is not exactly similar to that of the wild-type embryos. This may be due to the degeneration of ETX-structures. We removed Figure 7k according as suggested in the revised version.

I also suggest the text “AM” in Figure 7i in moved upwards/downwards, as it is now obstructing the ETX-structure. Again compare the place of the ETX-structure in Figure 7i (AM) with Figure 7k.

Response: Many thanks for your helpful suggestion, the text “AM” in Figure 7i was moved upwards accordingly. Please see figure 7i.

If the authors really want to show evidence that the ETX- structures resemble E5.5 embryo, I would expect they to provide not only staining for TE-derived cells, but also ICM-derived cells (Nanog/Oct4) and compare that nicely with natural embryos. As the evidence is lacking, I don't think the authors should claim so strongly that:

Line 359-360 “structures that resembled E5.5 embryos were detected in the decidual tissues at 48 h after transplantation”

Response: Thank you for your suggestion and apologies for any inadequate descriptions.

In fact, we detected E5.5 embryo-like structures in the decidual tissues triggered by the ETX-structures at 48 h after transplantation through H.E. staining. In the revised version, we added the detail information and cited the figures 7f and 7g at the end of the sentence.

Please see: “For transferred ETX-embryoids, E5.5 embryo-like structures were detected in the decidual tissues at 48 h after transplantation through H.E. staining (Fig. 7f, g)”.

--

Reviewer #2 (Remarks to the Author):

The revised manuscript by Zhang et al. describes a novel method for mimicking embryogenesis by assembling synthetic embryos from three different types of blastocyst-derived stem cells. This is an exciting approach which should be of interest to developmental biologists and stem cell researchers alike.

Although similar embryo-like structures have been described recently (Harrison et al, 2017), the ETX embryos generated here represent a significant advance due to the inclusion of all three cell lineages of the mouse blastocyst.

In their revision, the authors have addressed the major and minor concerns raised. They have provided additional data and quantifications to support their findings. Importantly, they have performed additional experiments to demonstrate that the emergence of DVE-AVE features progresses in a similar manner to natural embryogenesis and expanded their evidence for the initiation of implantation by ETX embryos. The accuracy of the language in the manuscript has also been improved. Together, these changes bring the paper up to a standard that warrants publication in Nature Communications. Finally, similar work has recently been published elsewhere, but I don't see this as a negative issue. The field will certainly benefit from multiple studies showing robustness and

reproducibility of this type of methods.

Response: Many thanks for your positive comment.